# Learning to Solve Constraint Satisfaction Problems with Recurrent Transformer

**Zhun Yang[1], Adam Ishay[1] & Joohyung Lee[1,2]**
[1]School of Computing and AI, Arizona State University, AZ, USA
[2]Global AI Center, Samsung Research, S. Korea
`{zyang90,aishay, joolee}@asu.edu`

## Abstract

Constraint satisfaction problems (CSPs) are about finding values of variables that satisfy the given constraints. We show that Transformer extended with recurrence is a viable approach to learning to solve CSPs in an end-to-end manner, having clear advantages over state-of-the-art methods such as Graph Neural Networks, SATNet, and some neuro-symbolic models. With the ability of Transformer to handle visual input, the proposed Recurrent Transformer can straightforwardly be applied to visual constraint reasoning problems while successfully addressing the symbol grounding problem. We also show how to leverage deductive knowledge of discrete constraints in the Transformer's inductive learning to achieve sample-efficient learning and semi-supervised learning for CSPs.

## 1 Introduction

Constraint Satisfaction Problems (CSPs) are about finding values of variables that satisfy given constraints. They have been widely studied in symbolic AI with an emphasis on designing efficient algorithms to deductively find solutions for explicitly stated constraints. In the recent deep learning-based approach, the focus is on inductively learning the constraints and solving them in an end-to-end manner. For example, the Recurrent Relational Network (RRN) (Palm et al., 2018) uses message passing over graph structures to learn logical constraints, achieving high accuracy in textual Sudoku. On the other hand, it uses hand-coded information about Sudoku constraints, namely, which variables are allowed to interact. Moreover, it is limited to textual input. SATNet (Wang et al., 2019) is a differentiable MAXSAT solver that can infer logical rules and can be integrated into DNNs. SATNet was shown to solve even visual Sudoku, where the input is a hand-written Sudoku board. The problem is harder because a model has to learn how to map visual inputs to symbolic digits without explicit supervision. However, Chang et al. (2020) observed a label leakage issue with the experiment; with proper evaluation, the performance of SATNet on visual Sudoku dropped to 0%. Moreover, SATNet evaluation is limited to easy puzzles, and SATNet does not perform well on hard puzzles that RRN could solve.

On another aspect, although these models could learn complicated constraints purely from data, in many cases, (part of) constraints are already known, and exploiting such deductive knowledge in inductive learning could be helpful for sample-efficient and robust learning. The problem is challenging, especially if the knowledge is in the form of *discrete* constraints, whereas standard deep learning is mainly about optimizing the *continuous* and *differentiable* parameters.

This paper provides a viable solution to the limitations of the above models based on the Transformer architecture. Transformer-based models have not been shown to be effective for CSPs despite their widespread applications in language (Vaswani et al., 2017; Zhang et al., 2020; Helwe et al., 2021; Li et al., 2020) and vision (Dosovitskiy et al., 2020; Gabeur et al., 2020). Creswell et al. (2022) asserted that Transformer-based large language models (LLMs) tend to perform poorly on multi-step logical reasoning problems. In the case of Sudoku, typical solving requires about 20 to 60 steps of reasoning. Despite the various ideas for prompting GPT-3, GPT-3 is not able to solve Sudoku. Nye et al. (2021) note that LLMs work well for system 1 intuitive thinking but not for system 2 logical thinking. Given the superiority of other models on CSPs, one might conclude that Transformers are unsuitable for CSPs.

We find that Transformer can be successfully applied to CSPs by incorporating recurrence, which encourages the Transformer model to apply multi-step reasoning similar to RRNs. Interestingly, this simple change already yields better results than the other models above and gives several other advantages. The learning is more robust than SATNet's. Looking at the learned attention matrices, we could interpret what the Transformer has learned. Intuitively, multi-head attention extracts distinct information about the problem structure. Adding more attention blocks and recurrences tends to make the model learn better. Analogously to the Vision Transformer (Dosovitskiy et al., 2020), our model can be easily extended to process visual input. Moreover, the model avoids the symbol grounding problem encountered by SATNet.

In addition, we present a way to inject discrete constraints into the Recurrent Transformer training, borrowing the idea from (Yang et al., 2022). That paper shows a way to encode logical constraints as a loss function and use Straight-Through Estimators (STE) (Courbariaux et al., 2015) to make discrete constraints meaningfully differentiable for gradient descent. We apply this idea to Recurrent Transformer with some modifications. We note that adding explicit constraint loss to all recurrent layers helps the Transformer learn more effectively. We also add a constraint loss to the attention matrix so that constraints can help learn better attentions. Including these constraint losses in training improves accuracy and lets the Transformer learn with fewer labeled data (semi-supervised learning).

In summary, the paper makes the following contributions. [1]

**Recurrent Transformer for Constraint Reasoning.** We show that Recurrent Transformer is a viable approach to learning to solve CSPs, with clear advantages over state-of-the-art methods, such as RRN and SATNet.

**Symbol Grounding with Recurrent Transformer.** With the ability of Transformers to handle vision problems well, we demonstrate that our model can straightforwardly be applied to visual constraint reasoning problems while successfully addressing the symbol grounding problem. It achieves 93.5% test accuracy on the SATNet's visual Sudoku test set, for which even the enhanced SATNet from (Topan et al., 2021) could achieve only 64.8% accuracy.

**Injecting Logical Constraints into Transformers.** We show how to inject discrete logical constraints into Recurrent Transformer training to achieve sample-efficient learning and semi-supervised learning for CSPs.

## 2 BACKGROUND

### 2.1 CONSTRAINT SATISFACTION PROBLEMS

A constraint satisfaction problem is defined as $\langle \mathbb{X}, \mathbb{D}, \mathbb{C} \rangle$ where $\mathbb{X} = \{\mathcal{X}_1, \ldots, \mathcal{X}_t\}$ is a set of $t$ logical variables; $\mathbb{D} = \{\mathbb{D}_1, \ldots, \mathbb{D}_t\}$ and each $\mathbb{D}_i$ is a finite set of domain values for logical variable $\mathcal{X}_i$; and $\mathbb{C}$ is a set of constraints. An *atom* (i.e., value assignment) is of the form $\mathcal{X}_i = v$ where $v \in \mathbb{D}_i$. A *constraint* on a sequence $\langle \mathcal{X}_i, \ldots, \mathcal{X}_j \rangle$ of variables is a mapping: $\mathbb{D}_i \times \cdots \times \mathbb{D}_j \rightarrow \{\text{TRUE}, \text{FALSE}\}$ that specifies the set of atoms that can or cannot hold at the same time. A (complete) *evaluation* is a set of $t$ atoms $\{\mathcal{X}_i = v \mid i \in \{1, \ldots, t\}, v \in \mathbb{D}_i\}$. An evaluation is a *solution* if it does not violate any constraint in $\mathbb{C}$, i.e., it makes all constraints TRUE.

One of the commonly used constraints is the *cardinality constraint*:

$$l \leq |\{\mathcal{X}_i = v_i, \ldots, \mathcal{X}_j = v_j\}| \leq u \tag{1}$$

where $l$ and $u$ are nonnegative integers denoting bounds, and for $k \in \{i, \ldots, j\}$, $\mathcal{X}_k \in \mathbb{X}$ and $v_k \in \mathbb{D}_k$. Cardinality constraint (1) is TRUE iff the number of atoms that are true in it is between $l$ and $u$. If $l = u$, constraint (1) can be simplified to

$$|\{\mathcal{X}_i = v_i, \ldots, \mathcal{X}_j = v_j\}| = l \tag{2}$$

which is TRUE iff the number of atoms in the given set is exactly $l$. If $i = j$ and $l = 1$, constraint (2) can be further simplified to $\mathcal{X}_i = v_i$ .

**Example 1 (CSP for Sudoku)** *A CSP for a Sudoku puzzle is such that* $\mathbb{X} = \{cell_1, \ldots, cell_{81}\}$ *denotes all 81 cells on a Sudoku board;* $\mathbb{D} = \{\mathbb{D}_1, \ldots, \mathbb{D}_{81}\}$ *and* $\mathbb{D}_i = \{1, \ldots, 9\}$ *(i = 1, \ldots, 81)*

---

[1]The code is available at `https://github.com/azreasoners/recurrent_transformer`.

*denotes all possible values in each cell; and $\mathbb{C}$ consists of constraints $cell_i = d$ for each given digit $d$ in cell $i$, and constraints*

$$|\{cell_i = d, \dots, cell_j = d\}| = 1 \tag{3}$$

*for $d \in \{1, \dots, 9\}$ and any set $\{i, \dots, j\}$ of 9 cell indices that belong to the same row/column/box, saying that "each digit $d$ should appear exactly once in each row/column/box." The solution to the CSP corresponds to the solution to the Sudoku puzzle.*

## 2.2 RELATED MODELS

**Graph Neural Networks (GNNs).** GNNs (Gori et al., 2005; Veličković et al., 2018; Kipf & Welling, 2017) are closely related to Transformers. They encode graph structures where adjacent nodes affect each other by recurrent message passing. The vanilla Transformer does not have an explicit encoding of the graph structure. In other words, it assumes fully connected graphs and does not exploit the sparsity of graphs. There have been many recent works to bring about the complementary nature of Transformers and GNNs, such as (Dehghani et al., 2018; Dai et al., 2019; Veličković et al., 2018; Yun et al., 2019; Rong et al., 2020; Cai & Lam, 2020; Hu et al., 2020; Ying et al., 2021; Dwivedi & Bresson, 2021).

**SATNet.** SATNet (Wang et al., 2019) explores semi-definite program relaxations as a tool for solving MAXSAT, which can be employed as a layer in deep neural networks to solve composite learning problems, such as visual Sudoku puzzles, that require both visual perception and logical reasoning. SATNet learns to solve visual Sudoku puzzles without any hand-coded knowledge, but its training relied on the "leakage" of labels, as discovered by Chang et al. (2020) and remedied by Topan et al. (2021). The following figure is from (Topan et al., 2021) to illustrate the *symbol grounding problem* in the context of a $3 \times 3$ portion of Sudoku. The task is to identify $a^{in}$ given only $a^{in}_{visual}$ and $a^{out}$ as the labels (called *Ungrounded* Dataset). However, the original SATNet training was performed on *Grounded* Dataset, where the labels also included $a^{in}$ so that a digit classifier was trained with the labels in a supervised way. Obviously, learning from the Ungrounded Dataset is harder because learning cannot be broken into two stages, classifying and solving.

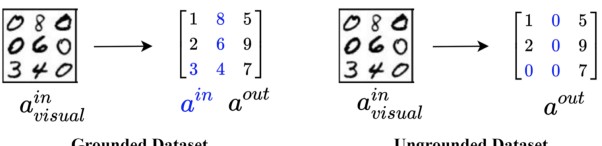

**Grounded Dataset**          **Ungrounded Dataset**

**Neuro-Symbolic Models for Constraint Reasoning.** CLR-DRNet (Bai et al., 2021) is a curriculum-learning-with-restarts model which leverages a Deep Reasoning Network (DRNet) (Chen et al., 2019) and is applied to visual Sudoku with a grounded dataset using rules given as a loss function but cannot be applied to Ungrounded Datasets. Neuro-symbolic models such as DeepProbLog (Manhaeve et al., 2018), NeurASP (Yang et al., 2020), and NeuroLog (Tsamoura et al., 2021) integrate neural networks with logic programming languages. They perform perception in neural networks and logical reasoning in hand-written logic programs. These models have shown that neural network training can benefit from constraints in logic programs.

# 3 RECURRENT TRANSFORMER FOR CONSTRAINT SATISFACTION PROBLEMS

## 3.1 RECURRENT TRANSFORMERS

Given a constraint satisfaction problem $\langle \mathbb{X}, \mathbb{D}, \mathbb{C} \rangle$ such that, for $i \in \{1, \dots, t\}$, $1 \leq |\mathbb{D}_i| \leq c$ for a constant $c$, Recurrent Transformer takes as input the sequence $\langle \mathcal{X}_1, \dots, \mathcal{X}_t \rangle$ of logical variables and outputs the probability distribution over the values in the domain $\mathbb{D}_i$ of each $\mathcal{X}_i$. Let $c_i$ be the domain size of $\mathcal{X}_i$. Without loss of generality, we assume that the values in $\mathbb{D}_i$ are represented by their indices, i.e., $\mathbb{D}_i = \{1, \dots, c_i\}$. The probability of $\mathcal{X}_i = j$ is given by the $j$-th value of the output for $\mathcal{X}_i$.

A logical variable $\mathcal{X}_i$ is treated as a token whose token embedding is a vector of length $d_h$ encoding the given information about this logical variable (e.g., some numbers, a textual description, an image, etc.). The positional embedding of $\mathcal{X}_i$ is a randomly initialized vector of length $d_h$ and is to be learned to record data-invariant information for logical variable $\mathcal{X}_i$. Let $\boldsymbol{E}_{tok}, \boldsymbol{E}_{pos} \in \mathbb{R}^{t \times d_h}$ denote the token and positional embeddings of $t$ logical variables. The $r$-th recurrent step in a Recurrent Transformer

with $L$ self-attention blocks and $R$ recurrences can be formulated as follows ($r \in \{1, \ldots, R\}$):

$$\boldsymbol{H}^{(r,0)} = \boldsymbol{H}^{(r-1,L)}$$
$$\boldsymbol{H}^{(r,l)} = \mathrm{block}_l(\boldsymbol{H}^{(r,l-1)}) \qquad\qquad \forall l \in \{1, \ldots, L\}$$
$$\boldsymbol{X}^{(r,l)} = \mathrm{softmax}(\mathrm{layer\_norm}(\boldsymbol{H}^{(r,l)}) \cdot \boldsymbol{W}_{out}) \qquad\qquad \forall l \in \{1, \ldots, L\}$$

where the initial hidden embedding $\boldsymbol{H}^{(0,L)}$ is $\boldsymbol{E}_{tok} + \boldsymbol{E}_{pos}$ and $+$ denotes element-wise addition; $\boldsymbol{H}^{(r,l)} \in \mathbb{R}^{t \times d_h}$ denotes the hidden embedding of $t$ logical variables after the $l$-th (self-attention) block in the $r$-th recurrent step; $\mathrm{block}_l$ denotes the $l$-th Transformer block in the model; $\mathrm{layer\_norm}$ denotes layer normalization; $\cdot$ denotes matrix multiplication, $\boldsymbol{W}_{out} \in \mathbb{R}^{d_h \times c}$ is the weight of the output layer; and $\boldsymbol{X}^{(r,l)} \in [0,1]^{t \times c}$ denotes the NN output with the hidden embedding $\boldsymbol{H}^{(r,l)}$. The figure of the model architecture and the formal definition of $\mathrm{block}_l$ are given in Appendix A.

For logical variable $\mathcal{X}_i$ and its domain $\mathbb{D}_i = \{1, \ldots, c_i\}$ where $c_i \leq c$, the scalar $X_{i,j}^{(r,l)}$ (i.e., element $i,j$ of matrix $\boldsymbol{X}^{(r,l)}$) is interpreted as the probability of atom $\mathcal{X}_i = j$ for $j \in \{1, \ldots, c_i\}$.

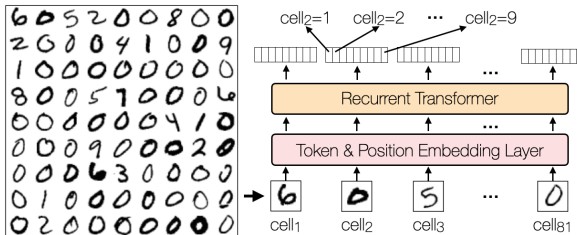

Figure 1: Recurrent Transformer for visual Sudoku problem.

**Example 2 (Recurrent Transformer for visual Sudoku)** *Figure 1 shows how Recurrent Transformer is used to solve the visual Sudoku problem from (Wang et al., 2019). Here, a Sudoku board is represented by $9 \times 9 = 81$ MNIST digit images where empty cells are represented by images of digit 0. The Recurrent Transformer takes as input the sequence $\langle cell_1, \ldots, cell_{81} \rangle$ of logical variables, and outputs the probability distribution over atoms $cell_i = v$ for $i \in \{1, \ldots, 81\}, v \in \{1, \ldots, 9\}$. The information given for each logical variable $cell_i$ is the MNIST digit image in the $i$-th cell. Within the Recurrent Transformer, the token, position, and hidden embeddings $\boldsymbol{E}_{tok}, \boldsymbol{E}_{pos}, \boldsymbol{H}^{(r,l)}$ are in $\mathbb{R}^{81 \times 128}$, and the output $\boldsymbol{X}^{(r,l)}$ is in $\mathbb{R}^{81 \times 9}$.*

**Recurrence in Transformers.** Adding recurrence to the standard Transformer is not a new idea, but its application to CSP is novel. Other implementations of recurrence in Transformers (Dehghani et al., 2018; Hao et al., 2019) apply causal attention to make the models predict the next token. In contrast, we use an encoder-only model with full attention and force the Transformer to update all unknown variables at every recurrence. Our Recurrent Transformer solves CSP problems incrementally, gradually predicting more unknown variables after it is confident in others. During the inference time, more recurrence steps can be used than those used in the training time, which can further boost performance. Furthermore, instead of computing a single loss on the final output as in Universal Transformer (Dehghani et al., 2018), we accumulate loss for each output from every attention block at every recurrent step, which yields better performance, as shown in Appendix B.2.

## 3.2 TRAINING OBJECTIVE

Consider a labeled data instance $\langle \boldsymbol{t}, \boldsymbol{l} \rangle$ where $\boldsymbol{t}$ is $t$ input tokens (which will be turned into the token and positional embeddings $\boldsymbol{E}_{tok}, \boldsymbol{E}_{pos}$) and $\boldsymbol{l} \in \{na, 1, \ldots, c\}^t$ is a label for $\boldsymbol{t}$, where $na$ denotes an unknown label. Let $\boldsymbol{X}^{(r,l)} \in \mathbb{R}^{t \times c}$ be the NN output with input $\boldsymbol{t}$ at recurrent step $r \in \{1, \ldots, R\}$ and block $l \in \{1, \ldots, L\}$. The cross-entropy loss $\mathcal{L}_{cross}$ is defined as follows, where $l_i$ denotes element $i$ in $\boldsymbol{l}$.

$$\mathcal{L}_{cross}(\boldsymbol{X}^{(r,l)}, \boldsymbol{l}) = - \sum_{i \in \{1,\ldots,t\}, \, j \in \{1,\ldots,c\}, \, l_i = j} log(X_{i,j}^{(r,l)}).$$

For example, in ungrounded visual Sudoku, $\boldsymbol{t}$ is a list of $t = 81$ MNIST images and $\boldsymbol{l} \in \{na, 1, \ldots, 9\}^{81}$ is the "ungrounded" solution for the Sudoku puzzle where the label for all given

digits is $na$. The cross-entropy loss on NN output $\boldsymbol{X}^{(r,l)} \in \mathbb{R}^{81 \times 9}$ depends only on the predictions of empty cells. In other words, no supervision for given digits is provided during training.

The baseline loss $\mathcal{L}_{base}$ is the sum of $\mathcal{L}_{cross}$ over the NN output $\boldsymbol{X}^{(r,l)}$ from all recurrent steps and blocks.

$$\mathcal{L}_{base} = \sum_{r \in \{1,...,R\},\ l \in \{1,...,L\}} \mathcal{L}_{cross}(\boldsymbol{X}^{(r,l)}, \boldsymbol{l}).$$

Note that we apply the cross-entropy loss to the NN outputs from all recurrent steps and all layers instead of from the very last one. We find that this makes the Recurrent Transformer converge faster.

## 4 EXPERIMENTS WITH RECURRENT TRANSFORMER

We use LxRyHz to denote our Recurrent Transformer with $L = x$ self-attention blocks, $R = y$ recurrent steps, and $z$ self-attention heads. If omitted, the number of heads $z$ is 4 and the embedding size $d_h$ is 128.

### 4.1 SUDOKU

In this section, we apply Recurrent Transformer to solve Sudoku problem, where a board can be either textual or visual.

**Dataset.** For textual Sudoku, we use the SATNet dataset from (Wang et al., 2019) and the RRN dataset from (Palm et al., 2018). The difference is that the RRN dataset is much harder and bigger (with 17-34 given digits in each puzzle and 180k/18k training/test data) than the SATNet dataset (with 31-42 given digits and 9k/1k training/test data). Each labeled data instance in textual Sudoku is $\langle \boldsymbol{t}, \boldsymbol{l} \rangle$ where $\boldsymbol{t} \in \{0,\ldots,9\}^{81}$ denotes a Sudoku puzzle (0 represents an empty cell) and $\boldsymbol{l} \in \{1,\ldots,9\}^{81}$ is the solution to the puzzle. For visual Sudoku, we use the ungrounded SATNet-V dataset from (Topan et al., 2021). SATNet-V was created based on the SATNet dataset where (i) each textual input in $\{0,\ldots,9\}$ in the training (or testing resp.) split is replaced with a randomly selected MNIST image in the MNIST training (or testing resp.) dataset, and (ii) the label for each given digit is $na$, i.e., unknown label that cannot be used to help training. In addition to SATNet-V, we created a new ungrounded dataset, RRN-V, following the same procedure based on the RRN data set. For faster evaluation of the RRN-V dataset, we randomly sampled 9k/1k training/test data and denoted it by "RRN-V (9k/1k)".

**Baselines and our Model.** We take RRN and SATNet as the baselines for textual Sudoku, and take RRN, SATNet, and SATNet* (Topan et al., 2021) (which resolves the symbol grounding issue of SATNet by clustering the input images) as the baselines for visual Sudoku. As RRN was not designed for visual Sudoku, we applied the same convolutional neural network (CNN) from (Wang et al., 2019) to turn each MNIST image into the initial number embedding of that cell in RRN. For our method on both textual and visual Sudoku, we apply Recurrent Transformer as in Figure 1 where the only difference is that the token embedding layer is a linear embedding layer for textual Sudoku and is the same CNN for visual Sudoku. All evaluations of our model use 32 recurrence steps for training and 64 for evaluation, as is done in (Palm et al., 2018).

Table 1: Whole board accuracy on different Sudoku datasets. RRN-hardest consists of a copy of the RRN training set, while the testing set consists of only the hardest puzzles with 17 given digits in the RRN test set.

| | | textual Sudoku | | | visual Sudoku (Ungrounded) | |
| | dataset | SATNet | RRN | RRN-hardest | SATNet-V | RRN-V |
| | #given | 31-42 | 17-34 | 17-34 | 31-42 | 17-34 |
| | (#train/#test) | (9k/1k) | (180k/18k) | (180k/1k) | (9k/1k) | (9k/1k) |
| Models | #Param (text/visual) | Accuracy on test data | | | | |
|---|---|---|---|---|---|---|
| RRN (Palm et al., 2018) | 201k / 692k | **100%** | 98.9% | 96.6% | 0% | 0% |
| SATNet (Wang et al., 2019) | 618k / 1049k | 98.3% | 6.1% | 0% | 0% | 0% |
| SATNet* (Topan et al., 2021) | – / 1049k + 13M(InfoGAN) | – | – | – | 64.8% | 0% |
| L1R32H4 (ours) | 211k / 702k | **100%** | **99.5%** | **96.7%** | **93.5%** | **75.6%** |

Table 1 shows that our method outperforms the state-of-the-art neural network models for both textual and visual Sudoku in different difficulties. Note that among all methods, only RRN requires prior

knowledge about Sudoku rules (i.e., there is an edge in the graph between every 2 nodes if their related cells are in the same row/column/box). Both RRN and SATNet fail on the (ungrounded) SATNet-V dataset due to the symbol grounding issue. While SATNet* could learn to solve visual Sudoku with the ungrounded dataset, it requires training an InfoGAN with 13M parameters to cluster the inputs. Unlike SATNet*, our model works out-of-the-box on visual Sudoku without carefully adjusting the structure and outperforms SATNet* by a large margin.

Although the L1R32H4 model has already achieved the new state-of-the-art results, we can further improve the accuracy by increasing the number $L$ of attention blocks, the number $H$ of heads, or the hidden embedding size $d_h$, with a trade-off of larger model size. We will analyze the effects of these decision choices on smaller datasets in the following sections.

### 4.1.1 ABLATION STUDY ON MODEL DESIGN (LxRyHz) WITH TEXTUAL SUDOKU

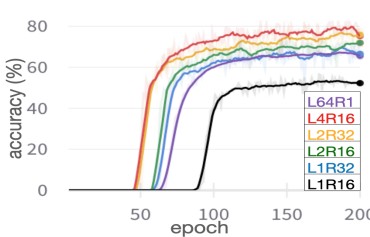 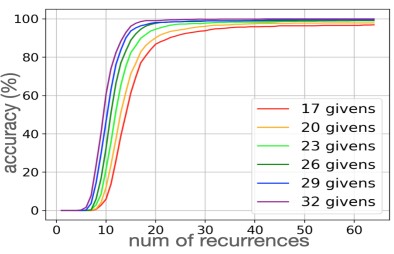

Figure 2: (**left**) Running average of the test accuracy for every 10 epochs of a Recurrent Transformer with different $L$ and $R$ trained on the same 8k RRN data. (**right**) Test accuracy as a function of the number $T$ of recurrences when testing on different difficulty puzzles in the RRN dataset, using the same L1R32 model trained on 180k RRN data.

**Effects of Blocks and Recurrences.** To analyze the effects of blocks and recurrences, we trained six LxRy(H4) models with different numbers of self-attention blocks $L$ and recurrences $R$ on an 8k/2k (training/test) RRN dataset with $\mathcal{L}_{base}$. Figure 2 (**left**) compares the whole board accuracy of these models, showing that more self-attention steps (equal to $L \times R$) lead to higher accuracy. With the same number of self-attention steps, when $L$ is small (e.g., $L \leq 4$), more parameters introduced by a larger $L$ slightly increase accuracy. On the other hand, adding recurrences is essential, and the non-recurrent model L64R1 performs poorly compared to the Recurrent Transformers.

The number of recurrences $T$ during testing can be higher than $R$ during training. Indeed, Figure 2 (**right**) shows that when $T \leq 64$, the more recurrent steps $T$ are, the higher accuracy is achieved with the same L1R32 model trained with 32 steps, and the improvement is bigger for harder puzzles.

**What Multi-Head Attentions Look at.** Without prior knowledge of the Sudoku game, Recurrent Transformer learns purely from $\langle puzzle, solution \rangle$ pairs that each cell should pay attention to all cells in the same row, column, and $3 \times 3$ box through the attention mechanism. We trained an L1R32 model with 1 to 4 self-attention heads on the SATNet dataset with $\mathcal{L}_{base}$. We found that the attentions on row, column, and box are clearly separated in different heads if the number of heads is greater or equal to 3 and would merge otherwise. Also, more attention heads help faster convergence, and accuracy may decrease if the number of attention heads is too small to capture different semantic meanings. More details and visualization of the attention matrices are given in Appendix B.1.

**Effect of Positional Embedding.** To evaluate the effect of positional embedding, we trained the L1R32 model without positional embedding on the SATNet dataset with $\mathcal{L}_{base}$, finding that removing positional embedding decreases the test accuracy from 100% to 0%. This is because positional embedding is essential for a CSP as it is the only source to differentiate logical variables (e.g., $cell_1, \ldots, cell_{81}$) with the same given information (e.g., digit 2 in both cell 4 and cell 10 in Figure 1).

### 4.1.2 ANALYSES ON SYMBOL GROUNDING WITH VISUAL SUDOKU

In visual Sudoku, we observed similar effects of different model designs as in textual Sudoku. We refer the reader to Appendix B for more details. In this section, we analyze how the symbol grounding issue is resolved in Recurrent Transformer by applying the same L1R32 model on both the RRN-V dataset and its grounded version (i.e., the label for every given digit is provided instead of $na$). For each of the two trained models, we evaluate their (i) whole board accuracy, (ii) *solution* accuracy where a board is counted correct if the prediction on all non-given cells is correct (even when the

given digits are incorrectly classified), and (iii) *givens cell* accuracy, i.e., the per-cell classification accuracy of the given cells.

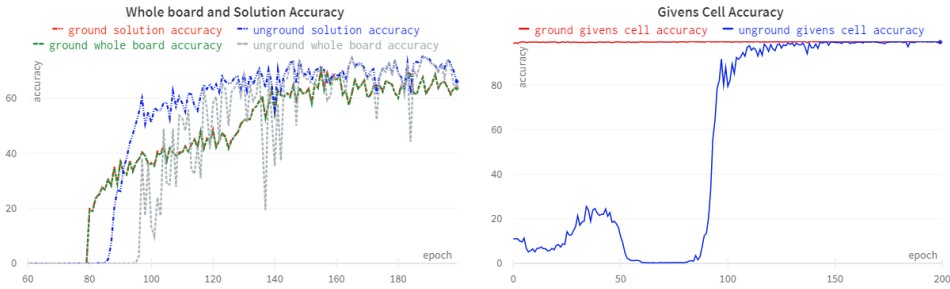

Figure 3: (**left**) The whole board accuracy and solution accuracy of the L1R32 model trained on the grounded or ungrounded RRN-V dataset (9k/1k). (**right**) The givens cell accuracy of the same models.

When trained on the grounded dataset, the L1R32 model quickly learns to classify the givens in 1 epoch, as shown in Figure 3 (**right**). On the other hand, with the ungrounded dataset, the L1R32 model starts to classify the givens correctly at around epoch 85. In Figure 3 (**left**) and (**right**), the solution accuracy and givens cell accuracy (with the ungrounded dataset) increase around the same time, indicating that digit classification is being jointly learned with solving. Interestingly, our model achieves 99.36% classification (givens cell) accuracy without explicitly training for it. Furthermore, the solution accuracy (75.5%) is consistently higher than the whole board accuracy (74.8%) as shown in Figure 3 (**left**), meaning that even when givens are not correctly classified, the solution can still be attained. We attribute this to the fact that reasoning is on the latent space instead of classifying and solving in two steps, as SATNet does.

## 4.2 OTHER EXPERIMENTS

Due to lack of space, we summarize other experiments with Recurrent Transformer below and refer the reader to Appendix D for the details.

**16x16 Sudoku.** We train our L1R32H8 model ($d_h = 256$) on 16x16 textual Sudoku. We generate two 10k (9k/1k training/test split) datasets of difficulty "simple" with an average of 111 givens and "medium" with an average of 95 givens. With 64 recurrent steps during inference, we achieved 99.9% accuracy on both test sets, meaning that there is only one wrongly predicted board solution. Due to the absence of a 16x16 Sudoku generator that can produce fewer givens, we could not test on harder boards.

**MNIST Mapping.** The MNIST Mapping problem was proposed in (Chang et al., 2020) as a simple test for the symbol grounding problem. It requires learning a bijection that maps an image of an MNIST digit to one of the 10 symbolic digits. (Chang et al., 2020) shows that SATNet is sensitive to this task and often fails without delicate tuning. Our Recurrent Transformer achieves 99% accuracy.

**Nonograms.** Nonogram (`https://en.wikipedia.org/wiki/Nonogram`) is a game that consists of an initially empty $N \times N$ grid representing a binary image, where each cell must take on a value of 0 or 1. Each row and column have constraints that must be satisfied to complete the image successfully. A constraint for a row/column is a list of numbers, where each number corresponds to contiguous blocks of cells with value 1 for a row/column. The Recurrent Transformer L1R16H4 achieved 97.5% test accuracy on 7x7 grids and 78.3% on 15x15 grids.

## 5 INJECTING LOGICAL CONSTRAINTS IN RECURRENT TRANSFORMER

### 5.1 INJECTING GENERAL CARDINALITY CONSTRAINTS VIA STE

Although Recurrent Transformer can learn to solve CSPs purely from labeled data, we could inject the known constraints to help it learn with fewer labeled data. In this section, we follow the idea from CL-STE (Yang et al., 2022) and propose a lightweight constraint loss method for a special family of constraints in CSP, namely the cardinality constraint that restricts the number of atoms in a set that can hold at the same time.

The main idea of CL-STE is to use a binarization function B to turn continuous values $x$ in NN output $\boldsymbol{X}^{(r,l)}$ into discrete values $\mathsf{B}(x)$ where $\mathsf{B}(x)$ is 1 (denoting TRUE) if $x \geq 0.5$ and 0 (denoting FALSE) otherwise. A constraint loss is then defined on these Boolean values and a set of propositional formulas in the Conjunctive Normal Form (CNF). Since $\frac{\partial \mathsf{B}(x)}{\partial x}$ is zero almost everywhere, the idea of STE is to replace $\frac{\partial \mathsf{B}(x)}{\partial x}$ with a straight-through estimator $\frac{\partial s(x)}{\partial x}$ for some (sub)differentiable function $s(x)$ so that the constraint loss has a non-zero gradient on $\boldsymbol{X}^{(r,l)}$.

While CL-STE has successfully injected discrete constraints into NN training, representing cardinality constraints in CNF is tedious. On the other hand, we notice that the binarization function $\mathsf{B}(x)$ enables direct counting on discrete values. Under this observation, for the cardinality constraint

$$l \leq |\{\mathcal{X}_{i1} = v_1, \ldots, \mathcal{X}_{ik} = v_k\}| \leq u$$

we construct a vector $\boldsymbol{x} \in \mathbb{R}^k$ of probabilities of the atoms in the given set such that $x_j$ (i.e., element $j$ in $\boldsymbol{x}$) is the probability of $\mathcal{X}_{ij} = v_j$ for $j \in \{1, \ldots, k\}$, and design a constraint loss as follows:

$$\mathcal{L}_{[l,u]}(\boldsymbol{x}) = \mathbf{1}_{c(\boldsymbol{x})<l} \times (c(\boldsymbol{x}) - l)^2 + \mathbf{1}_{c(\boldsymbol{x})>u} \times (c(\boldsymbol{x}) - u)^2 \tag{4}$$

where scalar $c(\boldsymbol{x}) = \sum \mathsf{B}(\boldsymbol{x}) = \sum_j \mathsf{B}(x_j)$, and $\mathbf{1}_{condition}$ is 1 if condition is true, 0 otherwise.

Similarly, constraint $|\{\mathcal{X}_{i1} = v_1, \ldots, \mathcal{X}_{ik} = v_k\}| = n$ can be encoded in the following loss.

$$\mathcal{L}_{[n]}(\boldsymbol{x}) = (c(\boldsymbol{x}) - n)^2. \tag{5}$$

In constraint losses (4) and (5), $c(\boldsymbol{x})$ is the number of 1s in the binarized vector $\mathsf{B}(\boldsymbol{x})$, which corresponds to counting the number of true atoms in constraints (1) and (2). Note that the binarization function $\mathsf{B}(x)$ enables the counting, but its gradient $\frac{\partial \mathsf{B}(x)}{\partial x}$ is always 0 whenever differentiable, so minimizing (4) and (5) will not work in updating NN parameters.

As with CL-STE, we use the identity STE to replace the gradient $\frac{\partial \mathsf{B}(x)}{\partial x}$ with 1 so that the gradient of each constraint loss to $\mathsf{B}(x)$ becomes the "straight-though estimator" of the gradient to $x$. In this way, we can do counting on the NN output with meaningful gradients. Although CL-STE could also represent the constraint loss (5) for $n = 1$ (uniqueness and existence of values), the size of the CNF representation could be huge.

**Example 3 (Constraint Loss on Output)** *Cardinality constraint loss (5) can be used to define the constraints in Sudoku problem*

$$\mathcal{L}_{Sudoku}(\boldsymbol{X}^{(r,l)}) = \sum_{k \in \{row,col,box\}} \sum_{i \in \{1,\ldots,81\}} \mathcal{L}_{[1]}(\boldsymbol{X}_{i,:}^k),$$

*where $\boldsymbol{X}^{(r,l)} \in \mathbb{R}^{81 \times 9}$ is the NN output; $\mathbf{X}^{row}, \mathbf{X}^{col}, \mathbf{X}^{box} \in \mathbb{R}^{81 \times 9}$ are reshaped copies of $\boldsymbol{X}^{(r,l)}$ such that each row in them contains the predictions in the same row/column/box; and $\boldsymbol{X}_{i,:}^k$ denotes row $i$ of matrix $\boldsymbol{X}^k$. Intuitively, $\mathcal{L}_{Sudoku}$ says that "exactly one digit in $\{1, \ldots, 9\}$ can be predicted in the same row/column/box". Note that, in CL-STE, the same Sudoku constraints are represented by a CNF with 729 atoms and 8991 clauses, which requires computation on a big matrix in $\{-1, 0, 1\}^{8991 \times 729}$.*

Furthermore, since the values in vector $\boldsymbol{x}$ are not limited to probabilities in NN outputs, we can apply these cardinality constraint losses to an attention matrix, representing additional constraints (not in the original CSP) that should be satisfied by the attention.

**Example 4 (Constraint Loss on Attention)** *In Sudoku problem, an attention matrix $\boldsymbol{A}^{(r,l)} \in \mathbb{R}^{81 \times 81}$ is computed in the $l$-th block at the $r$-th recurrence where $A_{i,j}^{(r,l)}$ is a normalized attention weight that can be interpreted as the percentage of attention from cell $i$ to cell $j$. The cardinality constraint loss (5) can also be used to define the following constraint loss*

$$\mathcal{L}_{attention}(\boldsymbol{A}^{(r,l)}) = \mathcal{L}_{[81]}(\boldsymbol{x})$$

*where $\boldsymbol{x} = \sum_j (\boldsymbol{A}_{:,j}^{(r,l)} \odot \boldsymbol{M}_{:,j})$; $\boldsymbol{M}$ is the adjacency matrix in $\{0, 1\}^{81 \times 81}$ such that $M_{i,j}$ is 1 iff cells $i$ and $j$ are in the same row, column, or box; $\odot$ denotes element-wise multiplication. Intuitively, the $i$-th element in $\boldsymbol{x} \in \mathbb{R}^{81}$ denotes the probability of the $i$-th cell paying attention to its adjacent cells. Minimizing $\mathcal{L}_{attention}(\boldsymbol{A}^{(r,l)})$ makes all 81 cells pay attention to their adjacent cells.*

Similarly to the baseline loss $\mathcal{L}_{base}$, which is the sum of $\mathcal{L}_{cross}$ over NN output $\boldsymbol{X}^{(r,l)}$ from all recurrent steps and blocks, the total constraint loss $\mathcal{L}_{constraint}$ is also accumulated over all NN outputs. The total loss with constraint loss is $\mathcal{L}_{total} = \mathcal{L}_{base} + \mathcal{L}_{constraint}$.

The constraint loss for the Sudoku problem is

$$\mathcal{L}_{constraint} = \sum_{r \in \{1,...,R\},\, l \in \{1,...,L\}} \Big( \alpha \mathcal{L}_{Sudoku}(\boldsymbol{X}^{(r,l)}) + \beta \mathcal{L}_{attention}(\boldsymbol{A}^{(r,l)}) \Big),$$

where $\alpha, \beta$ are reals in $[0, 1]$ that are hyper-parameters specified in Appendix F.2.

## 5.2 Experiments on Injecting Logical Constraints in Recurrent Transformer Training

Table 2: Effect of adding constraint losses $\mathcal{L}_{attention}$ (att) and $\mathcal{L}_{Sudoku}$ (sud) to the baseline loss $\mathcal{L}_{base}$ when training the same L1R32 model on 9k RRN or RRN-V training data.

| att | sud | textual Sudoku | | visual Sudoku | |
|---|---|---|---|---|---|
| | | $T$=32 | $T$=64 | $T$=32 | $T$=64 |
| – | – | 80.3% | 81.9% | 72.0% | 75.6% |
| – | ✓ | 80.1% | 84.4% | 74.4% | 79.3% |
| ✓ | – | 83.8% | 86.3% | 76.4% | 79.1% |
| ✓ | ✓ | 83.3% | 87.0% | 79.9% | 83.6% |

Table 3: Effect of adding constraint loss $\mathcal{L}_{constraint}$ and $x$ thousand **U**nlabeled data (denoted by $x$kU) when training the same L1R32 model on 4k **L**abeled RRN or RRN-V training data (denoted by 4kL).

| Data | textual Sudoku | | visual Sudoku | |
|---|---|---|---|---|
| | $T$=32 | $T$=64 | $T$=32 | $T$=64 |
| 4kL | 58.0% | 62.0% | 40.9% | 44.0% |
| 4kL + $\mathcal{L}_{constraint}$ | 65.4% | 69.2% | 47.5% | 50.4% |
| 4kL + 4kU + $\mathcal{L}_{constraint}$ | 65.8% | 69.9% | 57.2% | 61.0% |
| 4kL + 8kU + $\mathcal{L}_{constraint}$ | 70.7% | 73.3% | 60.8% | 64.4% |

To evaluate the effects of different logical constraint losses, we trained the L1R32 model on the 9k / 1k (training / test) RRN dataset and the 9k / 1k RRN-V dataset for 300 epochs until convergence with and without constraint losses. Table 2 shows that the same Recurrent Transformer model can further be improved if, in the total loss, we include $\mathcal{L}_{attention}$ and/or $\mathcal{L}_{Sudoku}$ on each neural network output, where the accuracy is evaluated with 32 or 64 recurrent steps $T$ during testing. We also observe a better performance gain with $\mathcal{L}_{Sudoku}$ than with $\mathcal{L}_{attention}$ because the baseline model (trained with $\mathcal{L}_{base}$ only) already learns the attention matrices well, as shown in Figure 5 in Appendix B.1. Besides, when we use 64 recurrences during testing (whereas trained with 32 recurrences), the same Recurrent Transformer model has bigger improvements in the test accuracy when it is also trained with constraint losses.

Since constraint loss $\mathcal{L}_{constraint}$ (accumulated by $\mathcal{L}_{Sudoku}$ and $\mathcal{L}_{attention}$) does not require labels, we could use it for semi-supervised learning tasks. Table 3 shows that, with only 4k labeled data, adding $\mathcal{L}_{constraint}$ increases the whole board accuracy of 1k test data, which can further be improved by adding additional 4k and 8k unlabeled data along with $\mathcal{L}_{constraint}$.

The shortest path problem is from (Xu et al., 2018) and is about finding the shortest path given a graph and the two endpoints. The example was used in (Xu et al., 2018; Yang et al., 2020) to demonstrate the effectiveness of semantic/constraint loss on neural network learning. Our experiment indicates that Recurrent Transformer achieves higher constraint accuracy, confirming the effects of the injected path constraints on Recurrent Transformer (Table 7 in Appendix D.1).

## 6 Conclusion

With the widespread success of Transformers on system 1 perception tasks, it is intriguing that they could also perform well on system 2 logical reasoning problems. Adding recurrences to the baseline model already outperforms the existing methods, especially on visual Sudoku puzzles with large margins (93.5% over enhanced SATNet's 64.8%), successfully addressing the issue of symbol grounding. We further improve the results by injecting underlying constraints into Transformer training so that the model can learn with fewer data, converge faster, and even improve accuracy. Our experiments show that more recurrences during training tend to yield higher test accuracy and additional recurrences during testing could also help. The number of attention blocks affects the size and modeling power of Recurrent Transformer. More attention heads lead to faster convergence, and the accuracy may decrease if the heads are too few to capture different semantic meanings.

ACKNOWLEDGEMENTS

This work was partially supported by the National Science Foundation under Grant IIS-2006747.

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

## A   RECURRENT TRANSFORMER DETAILS

Figure 4 shows a multi-layer Transformer encoder architecture (**a**) and the Recurrent Transformer architecture in our work (**b**), where every dotted box denotes a self-attention block. An output layer consists of a layer normalization, a linear layer, and a softmax activation function. In (**b**), all output layers share the same parameters, while every self-attention block has its own parameters.

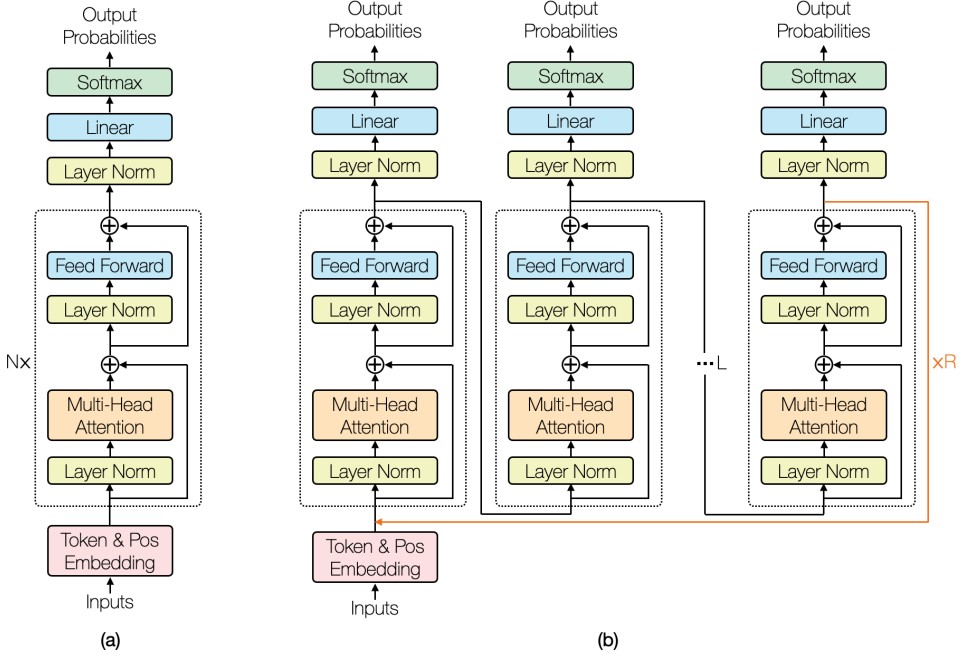

Figure 4: (**a**) Transformer encoder. (**b**) Recurrent Transformer encoder.

The Recurrent Transformer with $L$ self-attention blocks and $R$ recurrences can be formulated as follows:

$$\boldsymbol{H}^{(r,0)} = \boldsymbol{H}^{(r-1,L)} \qquad\qquad \forall r \in \{1,\dots,R\}$$
$$\boldsymbol{H}^{(r,l)} = \text{block}_l(\boldsymbol{H}^{(r,l-1)}) \qquad\qquad \forall r \in \{1,\dots,R\},\ \forall l \in \{1,\dots,L\}$$
$$\boldsymbol{X}^{(r,l)} = \text{softmax}(\text{layer\_norm}(\boldsymbol{H}^{(r,l)}) \cdot \boldsymbol{W}_{out}) \qquad \forall r \in \{1,\dots,R\},\ \forall l \in \{1,\dots,L\}$$

where $\boldsymbol{H}^{(r,l)} \in \mathbb{R}^{t \times d_h}$ denotes the hidden embeddings of $t$ input tokens after the $l$-th (self-attention) block in the $r$-th recurrent step, $block_l$ denotes the $l$-th Transformer block in the model (i.e., the $l$-th dotted box in Figure 4 (**b**)), layer\_norm denotes layer normalization, $\cdot$ denotes matrix multiplication, $\boldsymbol{W}_{out} \in \mathbb{R}^{d_h \times c}$ is the weight of the output layer for $c$ classes, and $\boldsymbol{X}^{(r,l)} \in [0,1]^{t \times c}$ denotes the NN output with the hidden embedding $\boldsymbol{H}^{(r,l)}$.

Each $block_l$ is defined on weights $\boldsymbol{W}_K^{(l)}, \boldsymbol{W}_Q^{(l)}, \boldsymbol{W}_V^{(l)}, \boldsymbol{W}_P^{(l)} \in \mathbb{R}^{d_h \times d_h}$ (for simplicity, we describe a single-head case) and a multilayer perceptron $\text{MLP}_l$ with output size $d_h$.

$$\mathbf{K}^{(r,l)} = \text{layer\_norm}(\boldsymbol{H}^{(r,l)}) \cdot \boldsymbol{W}_K^{(l)} \qquad\qquad \mathbf{Q}^{(r,l)} = \text{layer\_norm}(\boldsymbol{H}^{(r,l)}) \cdot \boldsymbol{W}_Q^{(l)}$$
$$\mathbf{V}^{(r,l)} = \text{layer\_norm}(\boldsymbol{H}^{(r,l)}) \cdot \boldsymbol{W}_V^{(l)} \qquad\qquad \mathbf{A}^{(r,l)} = \text{softmax}\left(\frac{\mathbf{Q}^{(r,l)}(\mathbf{K}^{(r,l)})^T}{\sqrt{d_h}}\right)$$
$$\mathbf{V}^* = (\mathbf{A}^{(r,l)} \cdot \mathbf{V}^{(r,l)}) \cdot \boldsymbol{W}_P^{(l)} + \boldsymbol{H}^{(r,l)}$$
$$\text{block}_l(\boldsymbol{H}^{(r,l)}) = \text{MLP}_l(\text{layer\_norm}(\mathbf{V}^*)) + \mathbf{V}^*$$

Here, $\boldsymbol{H}^{(r,l)}, \mathbf{K}^{(r,l)}, \mathbf{Q}^{(r,l)}, \mathbf{V}^{(r,l)}, \mathbf{V}^* \in \mathbb{R}^{t \times d_h}$ and $\mathbf{A}^{(r,l)} \in [0,1]^{t \times t}$.

The parameters are in terms of input vocabulary size ($v$), context size ($t$), number of classes ($c$), hidden embedding size ($d_h$), and the hidden layer size ($d_{MLP}$) of $\text{MLP}_l$, which is of shape ($d_h$, $d_{MLP}, d_h$).

(a) Parameter Values

| Parameter | Value |
|---|---|
| $v$ | 10 |
| $t$ | 81 |
| $c$ | 9 |
| $d_h$ | 128 |
| $d_{MLP}$ | 512 |

| Operation | Parameters | Parameter Count |
|---|---|---|
| Token Embedding | $v \times d_h$ | $10 \times 128 = 1280$ |
| Positional Embedding | $t \times d_h$ | $81 \times 128 = 10,368$ |
| Multi-Head Self-Attention $(\boldsymbol{W}_K^{(l)}, \boldsymbol{W}_Q^{(l)}, \boldsymbol{W}_V^{(l)}, \boldsymbol{W}_P^{(l)})$ | $4(d_h^2 + d_h)$ (the $d_h$ is for bias) | $4(128^2 + 128)$ $= 66,048$ |
| Layer normalization | $3 \times 2d_h$ | $3 \times 2 \times 128 = 768$ |
| $\text{MLP}_l$ $(d_h, d_{MLP}, d_h)$ | $d_h d_{MLP} + d_{MLP}$ $+ d_{MLP} d_h + d_h$ | $2 \times 128 \times 512 + 512$ $+128 = 131,712$ |
| Output layer $\boldsymbol{W}_{out}$ | $d_h c$ | $128 \times 9 = 1,152$ |

(b) Parameter Counts

Table 4: Parameter values and counts for L1R32H4 model for symbolic Sudoku.

As shown in Table 4, the parameters and their counts are shown. There are a total of 211,328 parameters. For SATNet(Wang et al., 2019), the number of parameters for Sudoku is 618,000 in total. This is $(n + 1 + aux) \times m$, where $n$ is the number of input variables, $aux$ is the number of auxiliary variables, and $m$ is the rank of the clause matrix. The (Palm et al., 2018) work has a total of 201,194 trainable parameters, which come from the row, column, and number embeddings, and the three MLPs used for node updates, message passing, and producing output probabilities.

## B MORE ABLATION STUDIES ON SUDOKU EXPERIMENTS

### B.1 EFFECTS AND VISUALIZATION OF MULTI-HEAD ATTENTION

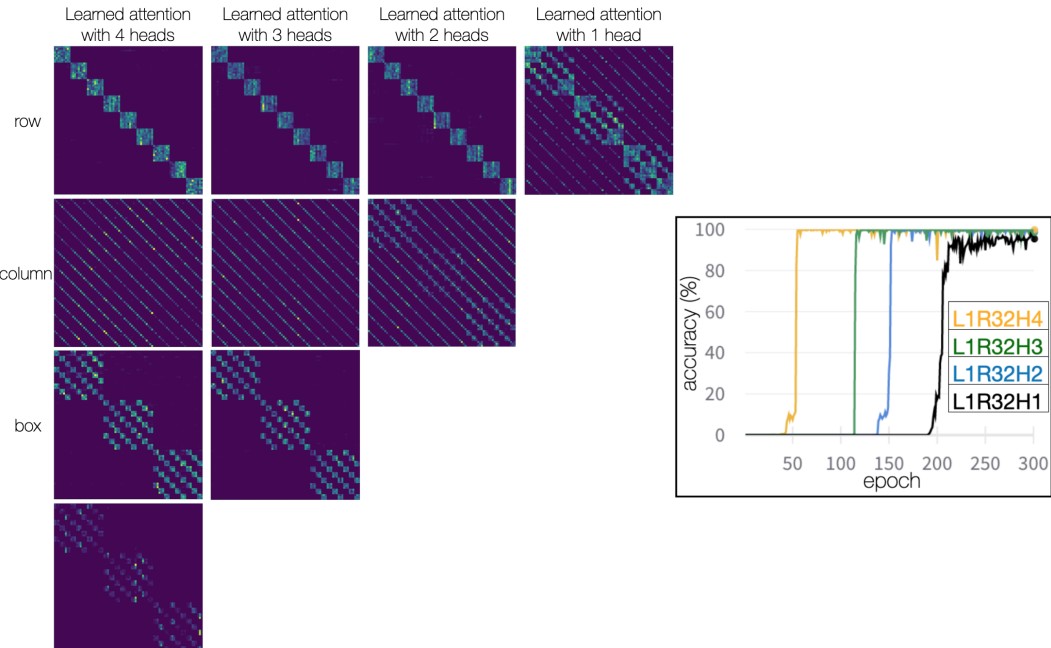

Figure 5: (**left**) Heatmaps of the learned 81x81 attention matrices in the L1R32 Recurrent Transformer with varying numbers of heads. (**right**) Test accuracy vs. epochs for these models.

Without prior knowledge of the Sudoku game, Recurrent Transformer learns purely from $\langle puzzle, solution \rangle$ pairs so that each cell should pay attention to all cells in the same row, column, and $3 \times 3$ box through the attention mechanism. We trained an L1R32 model with 1 to 4 self-attention heads on the SATNet dataset with $\mathcal{L}_{base}$. Figure 5 (**left**) visualizes the learned attention matrices – they correctly pay attention to each row, column, and box, respectively. For example, the first row of the top-left attention matrix in Figure 5 (**left**) learns purely from data about the 9 atoms to

pay attention in constraint (3) where $\{i,\ldots,j\} = \{1,\ldots,9\}$ and $d = 1$. These attentions are clearly separated into different heads if the number of heads is greater than or equal to 3 and would otherwise merge. Figure 5 (**right**) compares the whole board accuracy of these models, showing that more attention heads help accelerate convergence. The accuracy may decrease if the number of attention heads is too small to capture different semantic meanings.

## B.2 RECURRENT TRANSFORMER VS. VANILLA TRANSFORMER

There are two main decision choices in Recurrent Transformer: adding recurrence and applying losses to all blocks at all recurrent steps. To justify our decision choices, we compared 3 Transformer designs on the textual Sudoku problem under 3 settings. Figures 6, 7, and 8 show the experimental results on textual Sudoku on SATNet (9k/1k for training/testing), Palm (9k/1k), and Palm (3k/1k) datasets where

- the black line denotes the vanilla Transformer L32R1 with 32 blocks and with the cross-entropy loss applied to the final output;
- the yellow line denotes the Recurrent Transformer L1R32 with a single block, 32 recurrences, and with the cross-entropy loss applied to the last output;
- the red line denotes the Recurrent Transformer L1R32 with a single block, 32 recurrences, and with the cross-entropy loss applied to 32 outputs.

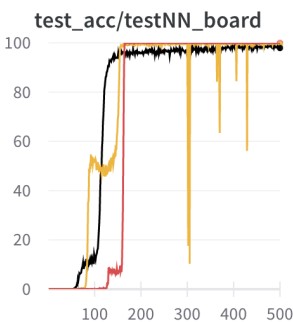

Figure 6: Whole-board test accuracy on SATNet (9k/1k)

Figure 7: Whole-board test accuracy on Palm (9k/1k)

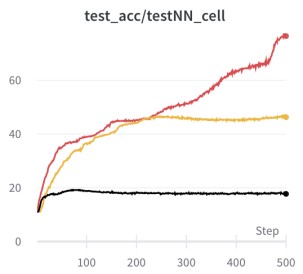

Figure 8: Cell test accuracy on Palm (3k/1k)

We can see that

- (comparing black and yellow lines) adding recurrences allows the model to achieve higher accuracy (especially for harder problems in the Palm dataset) with much fewer parameters in the model (1 block vs. 32 blocks);
- (comparing yellow and red lines) applying losses to all blocks makes the Recurrent Transformer model more stable and achieves higher accuracy than the Recurrent Transformer with a single loss;
- (comparing Figure 8 with the other 2 figures) the benefit of recurrence and losses on all blocks is greater when the number of data is smaller. Figure 8 compares the cell accuracy under the above 3 settings when trained on only 3k Palm data. In this figure, using recurrent blocks increases the converged cell accuracy from 17.7% to 46.3%, and applying losses to all blocks further improves the cell accuracy to 76.5%, and it has not converged.

## B.3 SEMI-SUPERVISED LEARNING WITH CONSTRAINT LOSS

In Table 3, we showed how constraint loss helps in a semi-supervised setting for textual and visual Sudoku. To analyze the effect of constraint loss on more unlabeled data instances, we continued the experiments for both textual and visual Sudoku and recorded the running average of the test accuracy for every 10 epochs in Figures 9 and 10.

We set the batch size to around 64 in the experiments in Figure 9 and to around 128 in the experiments in Figure 10. The batch sizes are slightly adjusted to have integer number split on labeled and

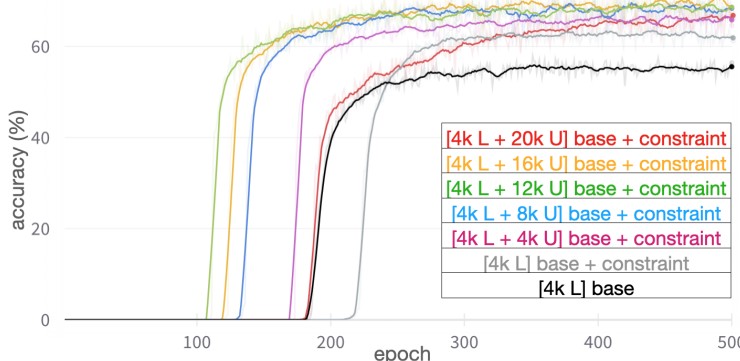

Figure 9: Effect of adding constraint loss $\mathcal{L}_{constraint}$ and $x$ thousand **U**nlabeled data (denoted by $x$k**U**) when training the same L1R32 model on 4k **L**abeled RRN training data (denoted by 4k**L**).

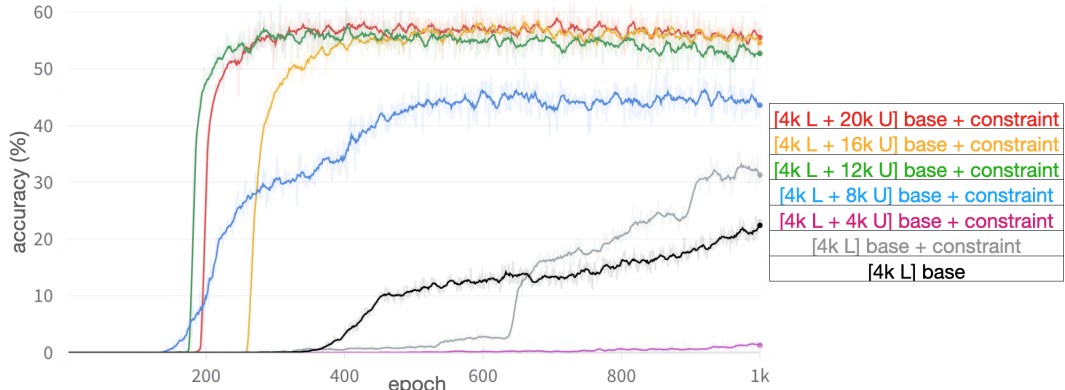

Figure 10: Effect of adding constraint loss $\mathcal{L}_{constraint}$ and $x$ thousand **U**nlabeled data (denoted by $x$k**U**) when training the same L1R32 model on 4k **L**abeled RRN-V training data (denoted by 4k**L**).

unlabeled data in each batch. To reduce the effect of hyper-parameter tuning on the weights of constraint losses, in all experiments, the weights $\langle \alpha, \beta \rangle$ of the constraint losses $\mathcal{L}_{sudoku}$ and $\mathcal{L}_{attention}$ are set to $\langle 1, 0 \rangle$, i.e., only the constraint loss $\mathcal{L}_{sudoku}$ is used with fixed weight 1. We can see that

- adding constraint loss improves the baseline accuracy by a large margin when trained with limited labeled data;

- with the help of constraint loss, adding more unlabeled data improves the accuracy in the beginning but the improvement is getting smaller;

- if we don't lower the weight for the constraint loss and keep increasing the number of unlabeled data, adding unlabeled data may lower the accuracy at some point as the signals (i.e., gradients) from constraint loss may overwrite the signals from the labels.

### B.4 CARDINALITY CONSTRAINT LOSS VS. CL-STE LOSS

The proposed cardinality constraint loss is different from the constraint loss in CL-STE Yang et al. (2022). We tried the original design of constraint loss in CL-STE but it computes too slowly due to the exponential size of CNF used to represent a cardinality constraint.

In Table 5, we applied the cross-entropy loss, the CL-STE loss, and the cardinality constraint loss to train the same RRN Palm et al. (2018) on the SATNet textual Sudoku dataset Wang et al. (2019). The cross-entropy loss serves as the baseline loss and is used in all four rows in Table 5 during training. Here, $R$ is the number of recurrent steps and is 32 in the RRN model; $NumAtom$ is the number of Boolean atoms in Sudoku and is $81 \times 9 = 729$; and $NumClause$ is 8991 which is the number of clauses in the CNF for Sudoku. As we can see, the proposed cardinality constraint loss has the same computation size as the cross-entropy loss, thus almost does not affect the training time. On

Table 5: Computation Size of Different Losses ($R = 32$, $NumAtom = 729$, $NumClause = 8991$)

| Loss | Applied To | Computation Size | Time/Epoch |
|------|------------|------------------|------------|
| Cross Entropy | all recurrent steps | $O(R \times NumAtom)$ | 120s |
| CL-STE | first recurrent step | $O(1 \times NumAtom \times NumClause)$ | 211s |
| CL-STE | all recurrent steps | $O(R \times NumAtom \times NumClause)$ | 3796s |
| Cardinality (ours) | all recurrent steps | $O(R \times NumAtom)$ | 122s |

the other hand, the constraint loss in CL-STE computes much slower since the computation size is propositional to the number of clauses in a CNF, whose size is exponential to represent a cardinality constraint. In addition to the cross-entropy loss that is applied to the output from all recurrent steps, if we only apply the CL-STE loss to the output from the first recurrent step as done in the CL-STE paper, the training time per epoch is 211s. Note that batch size is not included in the computation size for simplicity. All experiments use a batch size of 16 except for the third row (applying CL-STE loss to the outputs from all recurrent steps). If we apply the CL-STE loss to all recurrent steps, we have to decrease the batch size by 8 times to fit the GPU memory and the training time per epoch is increased to 3796s.

## C    DETAILED COMPARISON WITH SOTA MODELS FOR VISUAL SUDOKU

Topan et al. (2021) and Chang et al. (2020) showed that the original SATNet implementation failed on symbol grounding, which was not apparent due to a data leakage issue where labels for the input symbols were exposed during training time. Introduced in (Topan et al., 2021), the ungrounded visual Sudoku dataset does not include the symbolic label for the given numbers, while the grounded one does.

SATNet fails on ungrounded visual Sudoku (SATNet-V), achieving 0% accuracy. To address this, (Topan et al., 2021) alters SATNet by using InfoGAN to cluster the inputs and then jointly trains on MAXSAT to ground the symbols and solve Sudoku. They also slightly boost performance by using an additional linear "proofreading" layer, with a final ungrounded performance of 64.8% $\pm$ 3.0%. We train improved SATNet from (Topan et al., 2021) using the harder visual Sudoku dataset RRN-V (9k/1k), and it performs poorly, getting 0% whole board accuracy. Unlike the SATNet improved in (Topan et al., 2021), our model works out of the box on visual Sudoku without carefully adjusting the structure. Yet, it achieves 93.5% on the same dataset, as shown in Table 6. We also extended (Palm et al., 2018) to include an image embedding layer for visual Sudoku, denoted as RRN*. Although it performs well on textual Sudoku, it fails at symbol grounding for the same boards as images, achieving no better than random performance.

As described in (Topan et al., 2021), the SATNet paper (Wang et al., 2019) erroneously argued a performance bound of 74.8% ($0.992^{36.2}$) whole board accuracy based on the input classification accuracy of LeNet (99.2%) and the average number of givens (36.2). Our experiments confirm that this is wrong. We use the same CNN architecture that (Wang et al., 2019) uses, except that we change the output size to be the same as our embedding. Note that we achieved higher given cell accuracy with the same CNN (99.77% over 99.2%). Also, to the best of our knowledge, no simple CNN such as this achieves 99.77% accuracy on MNIST if trained for each image separately. Following the argument above, 92.0% ($= 0.9977^{36.2}$) would have been the theoretical maximum. However, our model achieves 93.5% whole board accuracy on the SATNet's visual Sudoku dataset, which indicates that digit classification learning considers the relational information of Sudoku.

Interestingly, our model's solution board accuracies are consistently higher than whole board accuracies, meaning that even when the givens are misclassified, the empty cells are filled with the right values. This is because our model does not wait to solve after the classification of givens is completed; rather, the classification and solving are jointly done, which also explains why the accuracy of the givens is on par with more advanced vision models, which use additional techniques like data augmentation, ensembling, etc.

Figure 3 compares our models trained with the grounded and the ungrounded versions of the RRN-V dataset. With the grounded dataset, the input labels are quickly learned, with the model achieving 99.46% givens cell accuracy just after the first epoch, while the ungrounded version takes significantly

Table 6: Results on visual Sudoku (ungrounded)

| Method | Dataset (#train/#test) | #givens | whole board acc. | solution board acc. | whole board cell acc. | solution cell acc. | givens cell acc. |
|---|---|---|---|---|---|---|---|
| SATNet (Wang et al., 2019) | SATNet-V (9k/1k) | 31-42 | 0% | – | 11.2% | – | 11.6% |
| SATNet* (Topan et al., 2021) | SATNet-V (9k/1k) | 31-42 | 64.8% | – | 98.4% | – | 98.9% |
| RRN* (Palm et al., 2018) | SATNet-V (9k/1k) | 31-42 | 0% | – | 11.56% | – | – |
| L1R32H4-V (ours) | SATNet-V (9k/1k) | 31-42 | 93.5% | 93.7% | 99.55% | 99.37% | 99.77% |
| SATNet (Wang et al., 2019) | RRN-V (9k/1k) | 17-34 | 0% | – | 11.63% | – | 12.70% |
| SATNet* (Topan et al., 2021) | RRN-V (9k/1k) | 17-34 | 0% | – | 31.08% | – | 74.03% |
| RRN* (Palm et al., 2018) | RRN-V (9k/1k) | 17-34 | 0% | – | 11.69% | – | – |
| L1R32H4-V (ours) | RRN-V (9k/1k) | 17-34 | 74.8% | 75.5% | 94.66% | 92.49% | 99.36% |
| L1R32H8-V (ours) | RRN-V (180k/18k) | 17-34 | 89.52% | 89.56 | 97.50% | 96.64% | 99.36% |

SATNet* indicates the altered SATNet from (Topan et al., 2021) and RRN* indicates RRN with an image embedding layer for visual Sudoku compatibility. Givens cell accuracy refers to the per-cell classification accuracy of the given digits. solution cell accuracy refers to the per-cell accuracy of solution cells (initially empty). For solution board accuracy, a board is counted as correct when all solution cells are correct. Whole board cell accuracy refers to the combined per-cell accuracy of both given and solution cells. For whole board accuracy, a board is counted as correct when all givens and empty cells are correct.

longer, achieving 99.36% after 143 epochs. The grounded dataset appears to help improve the whole board accuracy at first but converges at a lower solution board accuracy and whole board accuracy than the model trained with the ungrounded dataset. With the ungrounded dataset, the model starts to learn the givens and solutions at the same time, at around 85 epochs. This indicates that digit classification is being jointly learned during the solving process. However, as shown in Figure 3 (**left**), the model has a higher solution board accuracy than the whole board accuracy. This behavior is not surprising since it is enough to solve the non-given cells by just having a sufficient embedding of the given cells without classifying them. What is more interesting is that even though the model trained on the ungrounded dataset has no access to classification labels, it still learns to classify over 99% of the givens. The large oscillations of whole board accuracy for the model trained on the unground dataset is due to classification errors, mostly due to a single given digit being misrecognized. The graph also shows that for the ungrounded version, the solution board accuracy is consistently higher than the whole board accuracy, meaning that the unground version could still solve correctly even when the classification of input digits is wrong. On the other hand, for the model learned with the grounded dataset, the solution board accuracy overlaps with the whole board accuracy, implying a dependency between the solution board accuracy and givens/classification cell accuracy. The difference in the final whole-board accuracy between grounded and ungrounded is because the grounded version must optimize the classification of the input symbols while the ungrounded version does not.

# D  MORE DETAILS ABOUT OTHER EXPERIMENTS

## D.1  SHORTEST PATH

**Shortest Path in CSP.** A shortest path problem can be viewed as a CSP where $\mathbb{X} = \{node_1, \ldots, node_m, edge_1, \ldots, edge_n\}$ denotes all $m$ nodes and $n$ edges in a graph; $\mathbb{D} = \{\mathbb{D}_1, \ldots, \mathbb{D}_{m+n}\}$ and $\mathbb{D}_i = \{\text{FALSE}, \text{TRUE}\}$; and $\mathbb{C}$ is the set of constraints specifying the two end nodes in the graph and that "the selected edges form a path between the end nodes with a minimum length". The goal is to find a solution of this CSP, which represents the solution of the shortest path problem. Here, $node_i = \text{TRUE}$ (or $edge_i = \text{TRUE}$ resp.) represents that node $i$ (or edge $i$ resp.) is in the shortest path. Let $n_1, n_2 \in \{1, \ldots, m\}$ denote the indices of the 2 end nodes. $\mathbb{C}$ contains constraints $node_{n_1} = \text{TRUE}$ and $node_{n_2} = \text{TRUE}$, the following constraint for the end nodes $i \in \{n_1, n_2\}$,

$$|\{edge_{i1} = \text{TRUE}, \ldots, edge_{ik} = \text{TRUE}\}| = 1 \qquad (6)$$

and the following constraint for non-end nodes $i \in \{1, \ldots, m\} \setminus \{n_1, n_2\}$ in the shortest path (given from the label),

$$|\{edge_{i1} = \text{TRUE}, \ldots, edge_{ik} = \text{TRUE}\}| = 2 \qquad (7)$$

where $\{edge_{i1}, \ldots, edge_{ik}\}$ are the edges connected to node $i$ in the graph. The first constraint says that "each end node should connect to exactly 1 edge in the path" and the second constraint says that "each non-end node in the path should connect to exactly 2 edges in the path".

**Dataset.** We use the shortest path dataset SP4 from (Xu et al., 2018) to illustrate our method where each graph is a $4 \times 4$ grid with $m = 16$ nodes and $n = 24$ edges. SP4 has 1610 data instances and,

as in (Xu et al., 2018), we split the dataset into 60%/20%/20% training/test/validation examples. In addition, we created a more challenging dataset SP12 where each graph is a $12 \times 12$ grid with $m = 144$ nodes and $n = 264$ edges. SP12 has 22k data instances, split into 20k/1k/1k training/test/validation examples. In each problem, two end nodes are randomly picked up, and $\frac{n}{3}$ edges are randomly removed to increase difficulty. A labeled data instance is $\langle t, l \rangle$ where $t \in \{0,1\}^{m+n}$ such that $t_i = 1$ denotes "node $i$ is a terminal node" when $i \leq m$, and denotes "edge '$i-m$' is not removed" when $i > m$; and $l \in \{0,1\}^n$ such that $l_i = 1$ denotes "edge $i$ is in the shortest path."

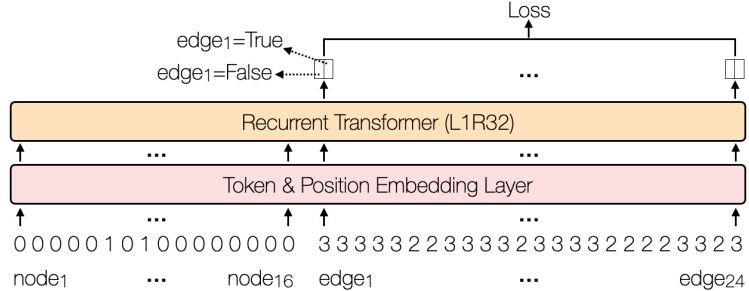

Figure 11: The representation used for the shortest path.

Figure 11 shows how Recurrent Transformer is used to solve the shortest path problem in a $4 \times 4$ grid with $m = 16$ nodes and $n = 24$ edges. The information given for each logical variable $node_i$ ($i \in \{1, \ldots, 16\}$) is a single digit 1 or 0 denoting that node $i$ is an end node or not. The given information for $edge_i$ ($i \in \{1, \ldots, 24\}$) is a single digit 2 or 3 denoting that the edge $i$ is removed or not. Given a NN output $\boldsymbol{X}^{(r,l)} \in \mathbb{R}^{40 \times 2}$ for 40 logical variables, we construct the vector $\boldsymbol{v} \in \mathbb{R}^{24}$ such that $v_i = X_{i+16,2}^{(r,l)}$, denoting the probabilities of $edge_i = \text{TRUE}$ for $i \in \{1, \ldots, 24\}$. Let $l \in \{0,1\}^{24}$ denote the label, i.e., $l_i = 1$ iff edge $i$ is in the shortest path. The cross-entropy loss is defined on $\boldsymbol{v}$ and $l$.

For the optional constraint loss, let $\boldsymbol{M}$ be the matrix in $\{0,1\}^{16 \times 24}$ such that $M_{i,j} = 1$ iff node $i$ is connected with edge $j$ in the graph. Let $\boldsymbol{c} \in \{0,1,2,3,4\}^{16}$ be $\boldsymbol{M} \cdot \boldsymbol{l}$. Intuitively, $c_i$ denotes the number of edges in the shortest path containing node $i$, and $c_i > 0$ means that node $i$ is in the shortest path. Then, constraints (6) and (7) can be encoded as follows:

$$\mathcal{L}_{path}(\boldsymbol{X}^{(r,l)}) = \sum_{i \in \{n_1, n_2\}} \left( \mathcal{L}_{[1]}(\boldsymbol{M}_{i,:} \odot \boldsymbol{v}) \right) +$$
$$\sum_{i \in \{1, \ldots, 16\} \setminus \{n_1, n_2\}} \left( \mathbf{1}_{c_i > 0} \times \mathcal{L}_{[2]}(\boldsymbol{M}_{i,:} \odot \boldsymbol{v}) \right),$$

where the non-zero values in $\boldsymbol{M}_{i,:} \odot \boldsymbol{v} \in \mathbb{R}^{24}$ are the probabilities of $edge_j = \text{TRUE}$ for all edge $j$ that contains node $i$.

Table 7: Constraint accuracy on SP4 test data for the shortest path problem

| Method | Constraint accuracy | | |
|---|---|---|---|
| | Path | No removed edges | Shortest path |
| MLP | 28.3% | 32.9% | 23.0% |
| MLP + Semantic Loss (Xu et al., 2018) | 69.9% | – | – |
| MLP + NeurASP (Yang et al., 2020) | 96.6% | 36.3% | 33.2% |
| L1R32 (ours) | 84.5% | 100% | 83.5% |
| L1R32 + $\mathcal{L}_{path}$ (ours) | 91.9% | 100% | 91.0% |

Table 7 compares the constraint accuracy achieved by (i) the baseline Multi-Layer Perceptron (MLP) introduced in (Xu et al., 2018) for the shortest path problem, (ii) the NeurASP method that encodes a path constraint to help train the MLP, (iii) our Recurrent Transformer (L1R32), and (iv) the same Recurrent Transformer enhanced by the constraint loss $\mathcal{L}_{path}$. The constraint accuracy are the percentage of the predictions that (i) form a valid path between end nodes, or (ii) do not include

removed edges, or (iii) form a shortest path between end nodes. Table 7 shows that Recurrent Transformer significantly outperforms the baseline MLP. Besides, the constraint loss $\mathcal{L}_{path}$ further improves the accuracy of the same Recurrent Transformer for predicting a valid path (or a shortest path resp.) from 84.5% (or 83.5%) to 91.9% (or 91.0%).

Furthermore, we applied the same L1R32 model to the more challenging SP12 dataset. After 2,000 epochs of training, we achieved 72.3% accuracy when trained with cross-entropy loss only and 76.0% when trained with both cross-entropy loss and constraint loss $\mathcal{L}_{path}$.

## D.2 Nonograms

Nonogram (`https://en.wikipedia.org/wiki/Nonogram`) is a game consisting of an initially empty $N \times N$ grid representing a binary image, where each cell must have a value of 0 or 1. Each row and column have constraints that must be satisfied to complete the image successfully. A constraint for a row/column is a list of numbers, where each number corresponds to contiguous blocks of cells for a row/column. We created two datasets for 7x7 and 15x15 grids, each having a 9k/1k training/test split. We use the same Recurrent Transformer model as in previous experiments. A sample $N \times N$ grid is input as a $N^2$ long sequence, where each element is a concatenated representation of the row and column constraints associated with the element. For example, for 15x15 grids, a given cell with column constraint [1,7,4] and row constraint [2,2,4,1] would have a corresponding sequence element of the concatenation of the two constraint vectors [0,0,1,7,4] and [0,2,2,4,1] (assuming the maximum constraint length is 5). With this simple input encoding only, our Recurrent Transformer L1R16H4 achieved 97.5% test accuracy on 7x7 grids and 78.3% test accuracy on 15x15 grids.

## E  Code & Datasets

Our Recurrent Transformer implementation is based on Andrej Karpathy's minGPT repository (https://github.com/karpathy/minGPT) under MIT license. We extend minGPT (a smaller version of GPT-3) to allow full attention, add recurrences, apply additional losses, and alter embeddings.

### E.1  Dataset Created by Others

**SATNet and SATNet-V Datasets.**  The SATNet and SATNet-V datasets are from SATNet (Wang et al., 2019) repository (`https://github.com/locuslab/SATNet`) under MIT license.

**RRN Dataset.**  The RRN dataset is from Recurrent Relational Networks (Palm et al., 2018) repository (`https://github.com/rasmusbergpalm/recurrent-relational-networks`) where no license information is provided.

**MNIST.**  We use MNIST images (LeCun et al., 1998) (`http://yann.lecun.com/exdb/mnist/`), which are available under the Creative Commons Attribution-Share Alike 3.0 license.

**SP4.**  We use $4 \times 4$ shortest path grids from (Xu et al., 2018) (`https://github.com/UCLA-StarAI/Semantic-Loss`), where no license information is provided.

### E.2  Datasets Created by Us

**RRN-V.**  The RRN-V dataset is created using the RRN dataset and MNIST images, similar to the way how SATNet-V dataset was created in SATNet repository. We construct the visual versions of RRN training/test datasets in which each board cell is represented by a randomly selected MNIST image.

**16x16 Sudoku.**  We generate 16x16 Sudoku boards using the generator here: `httdp://sudoku.smike.ru/hexsudoku.htm`, where no license is found. Though "medium" boards are actually labeled "hard" in their program, we label them "medium" since baseline performance is very high, and the percentage of givens is not low relative to the hardest 9x9 Sudoku hardest problems (17-givens, 21% of board).

**Nonograms.** For Nonograms dataset, we generate grids using the program "pattern.exe", available under the MIT license available here: `https://www.chiark.greenend.org.uk/~sgtatham/puzzles/`.

**SP12.** We generate $12 \times 12$ shortest path grids using the answer set solver CLINGO.[2]

## F  EXPERIMENTAL DETAILS

### F.1  COMPUTING

All of our experiments were done on Ubuntu 18.04.2 LTS with two 10-cores CPU Intel(R) Xeon(R) CPU E5-2640 v4 @ 2.40GHz and four GP104 [GeForce GTX 1080].

### F.2  TRAINING DETAILS

We used a fixed random seed $0$ for all experiments. The values of the weights $\alpha$ and $\beta$ of the constraint losses $\mathcal{L}_{sudoku}$ and $\mathcal{L}_{attention}$ are selected from $\{0, 0.1, 0.5, 1\}$ to achieve the highest training accuracy.

**Textual Sudoku.** For the weights $\langle \alpha, \beta \rangle$ of the constraint losses $\mathcal{L}_{sudoku}$ and $\mathcal{L}_{attention}$, we used $\langle 0, 0 \rangle$, $\langle 0, 1 \rangle$, $\langle 1, 0 \rangle$, and $\langle 0.5, 0.5 \rangle$ in the 4 textual Sudoku experiments in Table 2. The model structure and hyperparameters are shown in Table 8.

Table 8: Model Structure and Hyperparameters for Textual Sudoku Experiments

| Dataset | SATNET,RRN (9k/1k) | RRN (180k/(10k/1k)) |
|---|---|---|
| **Model Structure** | **Value** | **Value** |
| Number of attention heads | 4 | 8 |
| Number of layers | 1 | 1 |
| Number of recurrences (training/inference) | 32/64 | 32/64 |
| Embedding dimension | 128 | 256 |
| Token Embedder | Linear | Linear |
| Sequence length | 81 | 81 |
| **Hyperparameter** | **Value** | **Value** |
| Batch size | 16 | 16 |
| Learning rate | 6e-4 | 6e-5 |
| Dropout | 0.1 | 0.1 |

**visual Sudoku.** For the weights $\langle \alpha, \beta \rangle$ of the constraint losses $\mathcal{L}_{sudoku}$ and $\mathcal{L}_{attention}$, we used $\langle 0, 0 \rangle$, $\langle 0, 0.1 \rangle$, $\langle 1, 0 \rangle$, and $\langle 1, 0.1 \rangle$ in the 4 visual Sudoku experiments in Table 2. The model structure and hyperparameters are shown in Table 9.

**16x16 Sudoku.** The model structure and hyperparameters used for 16x16 Sudoku experiments are shown in Table 10.

**Shortest Path.** The model structure and hyperparameters used for shortest path experiments are shown in Table 11.

**MNIST Mapping.** The model structure and hyperparameters used for MNIST Mapping experiments are shown in Table 12.

**Nonograms.** A sample 7x7 Nonogram grid and solution is shown in Figure 12.

The model structure and hyperparameters used for Nonograms experiments are shown in Table 13.

---

[2]`https://github.com/potassco/guide/releases/tag/v2.2.0`

Table 9: Model Structure and Hyperparameters for visual Sudoku Experiments

| Model Structure | Value |
|---|---|
| Number of attention heads | 4 |
| Number of layers | 1 |
| Number of recurrences (training/inference) | 32/64 |
| Embedding dimension | 128 |
| Token Embedder | CNN |
| Sequence length | 81 |
| **Hyperparameter** | **Value** |
| Batch size | 16 |
| Learning rate | 6e-4 |
| Dropout | 0.1 |

Table 10: Model Structure and Hyperparameters for 16x16 Sudoku Experiments

| Model Structure | Value |
|---|---|
| Number of attention heads | 8 |
| Number of layers | 1 |
| Number of recurrences (training/inference) | 32/64 |
| Embedding dimension | 256 |
| Token Embedder | Linear Embedding |
| Sequence length | 256 |
| **Hyperparameter** | **Value** |
| Batch size | 24 |
| Learning rate | 6e-4 |
| Dropout | 0.1 |

Table 11: Model Structure and Hyperparameters for Shortest Path Experiments

| Dataset | 4x4 (10k/2k) | 12x12 (20k/1k) |
|---|---|---|
| **Model Structure** | **Value** | **Value** |
| Number of attention heads | 4 | 4 |
| Number of layers | 1 | 1 |
| Number of recurrences (training/inference) | 32/64 | 32/64 |
| Embedding dimension | 128 | 128 |
| Token Embedder | Linear | Linear |
| Sequence length | 40 | 408 |
| **Hyperparameter** | **Value** | **Value** |
| Batch size | 128 | 32 |
| Learning rate | 6e-4 | 6e-4 |
| Dropout | 0.1 | 0.1 |

Table 12: Model Structure and Hyperparameters for MNIST Mapping Experiments

| Model Structure | Value |
|---|---|
| Number of attention heads | 4 |
| Number of layers | 1 |
| Number of recurrences (training/inference) | 32/32 |
| Embedding dimension | 128 |
| Token Embedder | CNN |
| Sequence length | 1 |
| **Hyperparameter** | **Value** |
| Batch size | 256 |
| Learning rate | 6e-4 |
| Dropout | 0.1 |

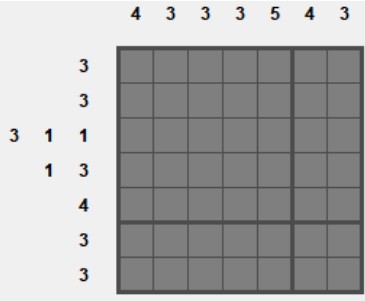 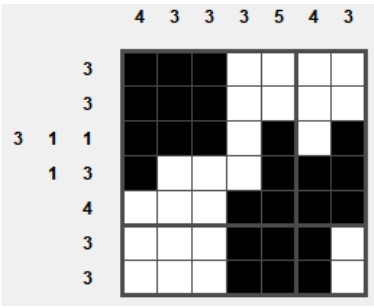

Figure 12: A sample 7x7 Nonogram grid (left) along with its solution (right).

Table 13: Model Structure and Hyperparameters for Nonograms Experiments

| Model Structure | Value |
| --- | --- |
| Number of attention heads | 4 |
| Number of layers | 1 |
| Number of recurrences (training/inference) | 16/32 |
| Embedding dimension | 128,320 (7x7,15x15) |
| Token Embedder | Linear Embedding |
| Sequence length | 49,225 (7x7,15x15) |

| Hyperparameter | Value |
| --- | --- |
| Batch size | 16 |
| Learning rate | 6e-4 |
| Dropout | 0.1 |

