# OpenReview forum: "Learning to Solve Constraint Satisfaction Problems with Recurrent Transformer"
_ICLR.cc/2023/Conference — ICLR 2023 poster_

### Official Review · Reviewer_SH5d · 2022-10-21

**Confidence:** 3
**Correctness:** 2
**Technical Novelty And Significance:** 2
**Empirical Novelty And Significance:** 2
**Recommendation:** 3

**Clarity, Quality, Novelty And Reproducibility:**

As recurrent transformers have already been proposed in the literature, the present architecture is not really novel. I would say that the main originality lies in the use of recurrent transformers for solving constraint satisfaction tasks. Yet, as indicated above, the overall architecture requires a lot of tuning and human expertise on injecting constraints as loss functions, and comparative experiments with standard constraint solvers are missing.

Finally, the paper has some clarity issues. In a constraint satisfaction problem, the overall task is to determine whether the instance is satisfiable, or not. When the CSP instance is satisfiable, a solution (i.e. complete assignment) should be provided. So, based on the architecture described in Section 3, and specifically the output layer $\mathbf{W}_{\mathit{out}}$, how do we extract a solution? Furthermore, what is given in output when the CSP instance is unsatisfiable?



**Strength And Weaknesses:**

As we know, constraint satisfaction problems have been a subject of extensive research for almost fifty years in AI. Nowadays, modern constraint solvers are able to efficiently solve a wide variety of constraint satisfaction tasks. So, the key question here is: why do we need another neural (or neuro-symbolic) architecture?

I would say that if the constraint satisfaction task involves some sub-symbolic components (images, etc.) then the use of a neural architecture is indeed relevant. To this very point, the experimental results reported in the paper indicate that recurrent transformers can handle visual Sudoku tasks in an end-to-end manner, which is a significant improvement with respect to existing neural models, such as RRN and SATNet.

On the other hand, for most constraint satisfaction tasks, the problem is symbolic in nature: it is defined using a set of discrete variables, a set of discrete domains, and a hypergraph of constraints. So, for those classical CSPs, there is no real need of using recurrent transformers, **unless they are in position to outperform standard solvers.**  From this perspective, the experiments should also include a comparison with standard solvers (e.g. FlatZinc, GeCode, Ace, etc.) on various CSP instances. Note here that the Shortest Path problem is easy (the problem is in P); I would recommend making comparisons on NP-hard tasks, such as, for example, resource allocation, scheduling or planning problems.

Beyond that, it seems that recurrent transformers require a lot of tuning in order to solve CSPs. Notably, the overall architecture involves up to 10 hyper-parameters, including the number of recurrences, the number of layers, the number of allocation heads, etc. Although recurrent transformers can learn to solve CSP instances using only labeled instances, their performance can be significantly improved by injecting logical constraints in the training objective (Sections 3.3 & 4.2). Yet here, we need to carefully design an appropriate loss function for each constraint type, which essentially means that the overall architecture is dedicated to the task at hand.



**Summary Of The Paper:**

In a constraint satisfaction problem (CSP), we are given a set of variables and a set of constraints. Each variable takes values in some discrete domain, and each constraint specifies the allowed combinations of values for some list of variables. The problem is to find an assignment of variables to values that satisfies all constraints. In this setting, the aim of this paper is to handle CSP instances using recurrent transformers. The overall architecture is defined using self-attention blocks, recurrent steps, and self-attention heads. Constraints can be injected using the CL-STE technique. Experiments are performed on visual 9x9 and 16x16 Sudoku tasks, shortest path problems, MNIST Mappings, and Nonogram puzzles.


**Summary Of The Review:**

To sum up, the main message of this paper is that recurrent transformers are amenable to “some” CSP tasks, with good performances in comparison with other neural architectures. Yet, I am not convinced that the paper can be accepted in its current state. Besides clarity issues, one important weakness is the lack of comparisons with standard solvers on various classical CSPs. Another weakness is the need for dedicated knowledge in order to adjust hyper-parameters and to inject constraints in the model.

---

> ### Author Response · Authors · 2022-11-13
> **Machine Learning Approach to CSP**
>
> Thank you for your valuable feedback. Below we address the concerns in your comments:
>
> > "modern constraint solvers are able to efficiently solve...why do we need another neural (or neuro-symbolic) architecture?"
> >  "unless they are in position to outperform standard solvers."
> >  “one important weakness is the lack of comparisons with standard solvers on various classical CSPs"
>
> This seems to be your main concern. You’re right that currently, all the works in neural net based CSP solvers (the award-winning SATNet paper [Wang et al, 2019], NeuroSAT [Selsam et al., 2019], etc.) are not at the scale to compete with standard CSP solvers. However, there is rising interest in the machine learning approach with different goals. We describe the benefits of this line of research (in the literature) as follows.
>
> 1. The machine learning approach enables learning constraints from <input, label> data pairs without specifying constraints, whereas the standard CSP solvers cannot. This is useful where expert knowledge is not sufficient or “too expensive to compute or mathematically not well defined”. For example, TSP in Montreal [Bengio et al., 2021] where customers vary every day and distances are only close to but not exactly l1 distance — solving this problem with an explicit definition is tedious and does not scale.
>
> 2. While it is not likely to solve general CSPs in poly-time (unless P=NP), a trained neural network model could serve as a fast approximation [Bengio et al., 2021], requiring poly-time in inference, at the expense of training hours.
>
> 3. It could “lead to improvements in practical SAT solving,” guiding “decisions within a more traditional SAT solver” [Selsam et al., 2018]. For example, [Selsam and Bjoner, 2019] trained a NeuroSAT architecture to predict the unsatisfiable cores of real problems directly.
>
> 4. It works on sub-symbolic domains like Visual Sudoku (as the reviewer noted)
>
> Thus, we think it’s not fair to judge our paper based on the “position to outperform standard solvers.” (which would be a reason to reject all above well-cited papers). Please note that our method outperforms those SOTA machine learning approaches in learning constraints from data by learning better with achieving higher accuracy. So, we kindly ask you to consider the above aspects in your evaluation.
>
> > "recurrent transformers require a lot of tuning...the overall architecture involves up to 10 hyper-parameters"
>
> Compared to a standard transformer-based model, the only new hyperparameter in our model is the number of recurrent steps. Sorry, the confusion about "too many hyperparameters" may come from the number of rows in Tables 8-12. We updated them in the revision following [Nandwani et al., 2022] and consider only batch size, learning rate, and dropout rate as hyperparameters. The other values are the settings for the model structure.
>
> > "how do we extract a solution?…what is given in output when the CSP instance is unsatisfiable?"
>
> The output of a recurrent transformer is a probability distribution of the values for all logical variables, thus is easily turned into a complete evaluation with argmax. Take Sudoku as an example. The final output X^{32,1} of a recurrent transformer model L1R32 is a matrix of probabilities in shape (81,9), and the final solution is a vector of shape (81) whose i-th element is in {1,...,9} and is the index of the element with the maximum value in the i-th row. This solution can be interpreted as a complete evaluation where "the i-th element being v" is interpreted as an atom "cell_i=v". Indeed, this is how we tested the accuracy.
>
> Like previous works [Rocktaschel and Riedel, 2017; Palm et al., 2018; Wang et al., 2019; Dong et al., 2019; Nandwani et al 2022], a recurrent transformer does not predict SAT/UNSAT but will always make a prediction even when there is no correct solution, which could be checked in poly-time.
>
> > "Another weakness is the need for dedicated knowledge in order to adjust hyper-parameters and to inject constraints in the model."
>
> Sorry about this confusion. First, compared to standard transformers, our model requires only one more hyperparameter — the number of recurrent steps. These hyperparameters are not fine-tuned and are almost the same across our experiments. Second, dedicated knowledge is not necessary to train a recurrent transformer. Table 1 shows even without them, our model outperforms SOTA models.  To avoid confusion, we restructured the paper and moved all constraint-related descriptions and experiments to Sec 5. Table 2 shows that we don't need the complete set of constraints.
>
> [Selsam et al., 2019] Learning a SAT solver from single-bit supervision
>
> [Bengio et al., 2021] Machine learning for combinatorial optimization: A methodological tour d’horizon
>
> [Selsam and Bjoner, 2019] Guiding High-Performance SAT Solvers with Unsat-Core Predictions
>
> [Rocktaschel and Riedel, 2017] End-to-end differentiable proving.
>
> [Dong et al., 2019] Neural Logic Machines

---

> > ### Author Response · Authors · 2022-11-18
> > **Follow up**
> >
> > Dear Reviewer SH5d,
> >
> > Since your concern is about the whole line of research (the machine learning approach to CSPs) rather than this particular submission, we're wondering if our answer is satisfactory to you. If not, please let us know so we have a chance to clarify further. Thank you!

---

### Official Review · Reviewer_wMfd · 2022-10-24

**Confidence:** 3
**Clarity, Quality, Novelty And Reproducibility:** The work presented is clear and seems…
**Correctness:** 2
**Technical Novelty And Significance:** 2
**Empirical Novelty And Significance:** 2
**Recommendation:** 3

**Strength And Weaknesses:**

I am confused about what is gained by adding recurrence to a transformer model. Large language models prior to transformers consisted of recurrent networks of various architectures that included attentional mechanisms, and can thus be described as recurrent transformers.

In my opinion, much of the gain found by the authors is likely due to the explicit constraint loss added to the intermediate layers, or due to injecting the logical constraints. In fact, it seems that the constraint loss on attention does a lot of the work of encoding the problem directly into the computation graph of the transformer.

**Summary Of The Paper:**

The authors consider the problem of learning to solve constraint satisfaction problems. This entails finding values of a set of variables that satisfy given constraints. The authors propose using a transformer model, extended with recurrence, to perform this variable assignment.

Existing work has shown that transformers produce subpar performance on constraint reasoning tasks, such as Sudoku. The authors incorporate recurrence into transformers to be able to successfully apply them to CSPs. Additionally, the authors add an explicit constraint loss to all recurrent layers.

**Summary Of The Review:**

My impression is that the authors do not fully understand the architecture they have built or why it performs the way it does. It seems to me that they have simply encoded the logical constraints of the domain into the architecture of the network itself. I don't think the recurrence has anything to do with the improved performance of the system.

---

> ### Author Response · Authors · 2022-11-13
> **Different kinds of recurrences for different purposes**
>
> Thank you for your valuable feedback. We are sorry about the confusion and appreciate the chance to clarify.
>
> > “I am confused about what is gained by adding recurrence to a transformer model. Large language models prior to transformers consisted of recurrent networks of various architectures that included attentional mechanisms, and can thus be described as recurrent transformers.”
>
> There seem to be two separate questions here.
>
> 1. the value of adding recurrence to transformer (which didn’t have recurrence as we do).
> 2. recurrent networks with attention mechanisms (but not transformers)
>
> Regarding 2, indeed, historically, there are other works using attention with RNNs [Bahdanau et al., 2014; Luong et al., 2015 ; Wu et al., 2016; Wang et al., 2016 ; Liu et al., 2018], but they are quite different from ours. First, RNNs at their core use recurrence as their primary mechanism, while attention is secondary if it is used. However, transformers have a core architecture that is primarily based on attention, therefore we won't call those RNNs with attention "transformers."
>
> Next, attention in transformers has been shown to replace the need for recurrence in RNNs. However, we are adding a different kind of recurrence. Recurrence in RNNs are "horizontally" applied to different, a sequence of inputs, while recurrence in our work is "vertically" applied to the same input (i.e., the same set of logical variables) repeatedly to update their embeddings.
>
> Regarding 1, please note that there are different forms of adding recurrence to a transformer model in the literature and they have shown to be useful for different purposes. For example,
>
> + [Dehghani et al., 2018] adds recurrence in both encoder and decoder of transformers so that the new model can generalize in, e.g., algorithmic tasks that recurrent models handle with ease.
> + [Hao et al., 2019] adds a recurrence encoder to transformers, which bridges the source and target sequences with a single recurrent layer and improves the performance on machine translation tasks.
> + [Dai et al., 2019; Hutchins et al., 2022] apply recurrent computation on windows of tokens to capture longer-term dependency beyond fixed-length context in long documents.
>
> In our paper, we also found it useful to add a form of recurrence for solving CSPs and empirically showed the benefit in Figure 2 (left), the discussion block "Effects of Blocks and Recurrences", and the ablation study in Appendix B.2.
>
>
> > "much of the gain found by the authors is likely due to the explicit constraint loss..."
>
> We are sorry about this confusion. Explicit constraint loss is not necessary to train a recurrent transformer, and all the results before section 4.2 do not use any constraint loss. Indeed, all models in Table 1 are trained on <input, output> data pairs only without using any constraint loss. That being said, our model is able to learn the constraints in Sudoku purely from ungrounded visual Sudoku data, and outperforms other SOTA approaches by a large margin. To address your concern due to the confusion, we restructured the paper as explained in the general response. We moved all constraint loss related descriptions and experiments to Section 5.
>
> > “it seems that the constraint loss on attention does a lot of the work of encoding the problem directly into the computation graph of the transformer.”
>
> It’s not true. Again, to avoid confusion, in the revision, we made Sec 3 and 4 without referring to constraint loss at all. Thus recurrent transformer learns constraints purely from data without constraint loss. Besides, the constraints losses on attention and output cannot directly encode the problem. We trained the same L1R32 model on SATNet textual dataset (9k/1k) with only the constraint losses (i.e., the cross-entropy loss is not added), and the performance is poor, with its cell accuracy fixed to be around 11%.
> The main message about constraint loss is that it augments the base cross-entropy loss, so that recurrent transformer converges faster using less labeled data as well as enables semi-supervised learning, but constraint loss alone doesn’t work.
>
> [Bahdanau et al., 2014] Neural Machine Translation by Jointly Learning to Align and Translate
>
> [Luong et al., 2015] Effective Approaches to Attention-based Neural Machine Translation
>
> [Wu et al., 2016] Google’s Neural Machine Translation System: Bridging the Gap between
> Human and Machine Translation
>
> [Wang et al., 2016] Attention-based LSTM for Aspect-level Sentiment Classification
>
> [Liu et al., 2018] Recurrent networks with attention and convolutional networks for sentence representation and classification
>
> [Dehghani et al., 2018] Universal transformers.
>
> [Hao et al., 2019] Modeling recurrence for transformer.
>
> [Dai et al., 2019] Transformer-XL: Attentive language models beyond a fixed-length context
>
> [Hutchins et al., 2022] Block-recurrent transformers

---

> > ### Author Response · Authors · 2022-11-18
> > **Follow up**
> >
> > Dear Reviewer wMfd,
> >
> > In our rebuttal, we addressed your question about adding recurrence to transformers. We also clarified that adding explicit constraint loss is auxiliary; even without it, our method already outperforms SOTA approaches. Please let us know if you still have questions. Thank you!

---

### Official Review · Reviewer_V9ce · 2022-10-25

**Confidence:** 5
**Correctness:** 3
**Technical Novelty And Significance:** 3
**Empirical Novelty And Significance:** 4
**Recommendation:** 8

**Clarity, Quality, Novelty And Reproducibility:**

The paper is well written. All the details of the experiments have been provided in the appendix, and the source code has also been shared, though I haven’t checked the code.

Novelty in terms of new ideas isn’t much, but the paper does a very good job at successfully bringing together different ideas for solving a task. As a result, it achieves s-o-t-a performance in a difficult reasoning task of ungrounded-visual sudoku. As pointed by the authors, the existing s-o-t-a framework for ungrounded visual sudoku requires additional training of InfoGANs, and on the other hand, the proposed architecture is elegant, end-to-end and doesn’t require any external components, such as InfoGANs for clustering.


**Strength And Weaknesses:**

As the authors have pointed out, the idea of recurrence in Transformers is not novel, but they have been able to successfully apply it to learn the constraints of a CSP.

The idea of computing loss using the output of every recurrent step has already been used to train RRNs.

Also, there is a lot of work on using Soft Logic to convert constraints on discrete symbols to equivalent constraints on the symbols’ probabilities computed by a neural network. For example, see Li et al. 2019.  Some of the formulations may give a similar loss term as the proposed constraint loss.

The task is to solve CSP by implicitly learning the underlying constraints in the network’s weights, but on the other hand, the constraint loss term assumes explicit knowledge of the same unknown constraints.  Though I agree that it is not clear how the knowledge of rules can be used with the ungrounded datasets, but for textual sudokus, if we know the constraints, we can create unlimited training data by using a symbolic solver. Maybe the authors should clarify this in the paper.  Also, the paper points out (almost criticizes) at a couple of places that RRN uses additional knowledge of which variables are allowed to interact, but the proposed constraint loss term uses almost similar knowledge.

Having said that, the paper does a good job of bringing together all of these ideas and proposing architecture for solving combinatorial problems.

It archives s-o-t-a performance on difficult ungrounded visual sudoku datasets, beating the baseline by a huge margin.

The authors have done extensive analysis and ablations to demonstrate the effectiveness and workings of the proposed approach.


[Li et al 2019] A Logic-Driven Framework for Consistency of Neural Models.


**Summary Of The Paper:**

The paper proposes a recurrent architecture using Transformers to solve instances of a Constraint Satisfaction Problem with unknown constraints.  Weight sharing between the recurrent transformer layers allows it to create a much deeper network, which is required for solving combinatorial problems like sudoku. It also adapts the formulation of CL-STE (Yang et al 2022) to convert cardinality constraints on Boolean variables to a loss function on the probabilities of those Boolean variables being true. It experiments on symbolic as well as visual sudokus and achieves state-of-the-art results on ‘ungrounded-visual sudoku’ in which output symbols for only the empty cells are given in the training data.

**Summary Of The Review:**

Overall, I like the paper for its clever use of existing ideas for demonstrating the effectiveness of transformers in learning constraints and solving combinatorial problems.  The experimentation is very thorough and extensive.
They achieve state-of-the-art performance on ungrounded visual sudoku and beat the baseline by a huge margin.


Question for the authors:
1.  For 16 x 16 sudokus, are the experiments on textual, or ungrounded visual sudokus? I couldn’t find this detail in the paper.
2. Table 1, L1R32H4 on RRN-V achieves 74.8% board accuracy. Is this using the constraint loss terms? Why don't we have this number in Table 2?
3. Did you try to train RRN with the constraint loss terms? I would encourage you to do so and report it in the paper.
4. "As RRN was not designed for Visual Sudoku..." (Pg 7): this claim is not true. Just because RRN authors didn't experiment with visual sudoku, it doesn't imply that RRN wasn't designed for it. As you have demonstrated, RRN can be easily plugged in as a reasoning component in a neural architecture.
5. Is it possible to extend transformers architecture to learn first-order rules instead of grounded CSPs, like NLM (Dong et al 2019) does? Specifically, can we extend recurrent transformers to train using 9 x 9 sudokus, but test on 16 x 16, like (Nandwani et al 2022) extend RRNs to do so? Given that positional embeddings are critical for training transformers, you may no longer be able to use fixed positional embeddings and may have to try something like ALiBi (Pres et al 2021), or relative positional embeddings (Shaw et al 2018).

[Dong et al 2019] Neural Logic Machines
[Nandwani et al 2022] Neural Models for Output-Space Invariance in Combinatorial Problems
[Shaw et al 2018] Self-Attention with Relative Position Representations

---

> ### Author Response · Authors · 2022-11-13
> **Thanks for constructive comments and suggestion for the future work**
>
> Thanks for reviewing our paper and providing your valuable feedback! We are glad you found our paper well written, the way to bring different ideas successful, the experimentation and analysis thorough and extensive, and the improvements on difficult reasoning tasks significant. We have updated our paper to reflect your comments. Please take a look at our general response for highlights of the revision. Below are our answers to your questions.
>
> 1. The experiments on 16x16 Sudoku are on textual Sudoku.
>
> 2. The accuracy 74.8% in Table 1 is the result when no constraint loss is used. The number is a typo (thanks!) and was updated to the accuracy 75.6% in the revision.
>
> 3. [Yang et al., 2022] analyzed the effect of applying constraint loss in CL-STE to train RRN for textual Sudoku problems. Compared to the constraint loss in CL-STE, the proposed cardinality constraint loss has a much smaller computation size. Please refer to bullet 3 in the general response for a more detailed comparison.
>
> 4. Thanks for pointing it out. We updated the statement to "As RRN wasn't applied to visual Sudoku" in the revision. As described in the paper, we applied RRN to visual Sudoku by using the same convolutional neural network (CNN) from (Wang et al., 2019) to turn each MNIST image into the initial number embedding of that cell in RRN. However, simply plug-in RRN as a reasoning component in a neural architecture in this way didn’t work, as we reported in Table 1.
>
> 5. Thank you for your valuable suggestions and references! Generalizing across the size of the output space (e.g., train on 9x9 Sudoku and test on 16x16 Sudoku) is an exciting and challenging topic. Different from a GNN where the connectivity of nodes is pre-defined for both 9x9 and 16x16 Sudoku, the connectivity in a transformers-based model needs to be learned in the attention matrix of size 81x81 or 256x256. Since the model of a CSP is invariant to the order/location of the logical variables, more thoughts about relative positional embedding are needed, although it seems to be a good fit for Sudoku problem (without permutation). Integrating ALiBi [Press et al., 2021] carefully may be a promising direction. For example, adding a mask to the attention which encodes the row, column, and box information for Sudoku. Therefore when generalizing to 16x16 Sudoku, the additional grid cells not seen in training will have information about which row, column, and box they belong to. In light of your comment, we have tried a variation of ALiBi on our recurrent transformer to achieve the same performance on the baseline without positional embeddings. With this working baseline we can work towards generalizing to larger CSP problems, perhaps borrowing some ideas from [Nandwani et al., 2022] to achieve value-set invariance. Given that [Nandwani et al. 2022] achieved only 27.31% accuracy on 16x16 Sudoku, we believe this challenge is harder than it appears and leaves further exploration in our future work.
>
> Finally, we thank the reviewer again for the thoughtful comments and constructive suggestions! We hope our revised version successfully addressed your concerns and questions!

---

### Official Review · Reviewer_KSwb · 2022-10-30

**Confidence:** 4
**Correctness:** 4
**Technical Novelty And Significance:** 2
**Empirical Novelty And Significance:** 3
**Recommendation:** 6

**Clarity, Quality, Novelty And Reproducibility:**

This paper is generally well-written, well-presented, and sufficiently well evaluated. My main concern here is with respect to novelty. At this point in time, transformers have been applied successfully to every problem known to machine learning and have become almost a universal tool for solving anything. In this light it's not a big surprise that this particular set of applications yields good results. The techniques that handle the input/output encoding are not new either by the author's own admission. On the flip side, based on results alone, I imagine that this paper would be of interest to some fraction of the community since it fixes known problems in the state of the art.

**Strength And Weaknesses:**

Strengths:
* the presented architecture is fairly generic and applicable to many different types of input and problem structures;
* the results show substantial improvements over existing algorithms;
* generally well-written and reasonably evaluated;

Weaknesses:
* the novelty is not particularly high;

**Summary Of The Paper:**

This paper proposes an architecture for solving CSPs based on a transformer network. The transformer makes multiple passes over the inputs, keeping information across different passes in the form of a recurrent state. The network is also augmented with a constraint loss that injects problem knowledge into the training process. The results are evaluated on visual Sudoku as well as other benchmarks showing substantially improved results.

**Summary Of The Review:**

An application paper where known techniques are combined to yield substantial improvements over SOTA. Low novelty is somewhat offset by results, so there is an argument that this should be published.

---

> ### Author Response · Authors · 2022-11-13
> **New ideas on the design to make it work and new constraint loss**
>
> We are glad you found our method generic and well-evaluated, our improvements over SOTA substantial, and our paper generally well-written. Although transformers are widely used in many other areas, their capability on CSPs has not been explored. Also, our experience is that we had to come up with a few new methods to make it work on CSPs, as explained below.
>
> - Although recurrence in transformers is not new, previous works [Dehghani et al., 2018; Hao et al., 2019; Dai et al., 2019; Hutchins et al., 2022] focus on sequence-to-sequence prediction and use a causal attention to encourage the model to focus on the next token to be generated. This is not suitable for CSPs where the value of logical variables may be predicted in any order. We use an encoder-only model with full attention to force the model to update all unknown variables at every recurrence. This allows the model to produce confident predictions at any point and in any order, updating with each pass until a final solution is produced. Additionally, each token in previous works is aligned with a single loss while we align each token with R losses to encourage the model to solve a CSP incrementally and learn a convergent reasoning algorithm in each recurrence. This is unusual but is essential when the problem becomes hard, as shown in our ablation study in Appendix B.2.
>
> - The constraint loss we designed is different from that in CL-STE [Yang et al., 2022]. We tried the original design of constraint loss in CL-STE, but it computes too slow due to the exponential size of CNF used to represent a cardinality constraint. Thus, we invented a new constraint loss for a general cardinality constraint based on counting discrete values in either NN output or attention matrix, which, to the best of our knowledge, is new. This is documented in Appendix B.4 in the revision. A snapshot of the comparison is shown in the following table
>
> | Loss      |         Applied_To | Computation_Size  |         Time/Epoch
> |-----------|--------------------| -------------------| ------------|
> |	Cross_Entropy   |   All_Rec |   O(R x NumAtom)   |          120s |
> |	CL-STE (original) | 1st_Rec |   O(1 x NumAtom x NumClause)| 211s  |
> |	CL-STE (variant)  | All_Rec  |  O(R x NumAtom x NumClause) |3796s  |
> |	Cardinality (ours) | All_Rec  |  O(R x NumAtom)             |122s|
>
> All_Rec means the loss is applied to the output from all recurrent steps; 1st_Rec means the loss is only applied to the output from the first recurrent step (which is the setting in CL-STE); NumAtom is 7x81=729 and NumClause is 8991 in Sudoku; and Time/Epoch is the training time per epoch. The cross-entropy loss serves as the baseline loss and is used in all four rows during training. Note that in CL-STE paper, the constraint loss is only applied to the output from the first recurrent step while, in our paper, we need to apply the constraint loss to the output from all recurrent steps. “CL-STE (variant)” denotes the new setting where we apply the constraint loss in CL-STE to all recurrent steps. As we can see, the proposed cardinality constraint loss has the same computation size as the cross-entropy loss, thus almost doesn't affect the training time. On the other hand, the constraint loss in CL-STE computes much slower since the computation size is propositional to the number of clauses in a CNF, whose size (8991) is exponential to represent a cardinality constraint.
>
>
> [Dehghani et al., 2018] Universal transformers
>
> [Hao et al., 2019] Modeling recurrence for transformer
>
> [Dai et al., 2019] Transformer-XL: Attentive language models beyond a fixed-length context
>
> [Hutchins et al., 2022] Block-recurrent transformers
>
> [Yang et al., 2022] Injecting Logical Constraints into Neural Networks via Straight-Through Estimators

---

### Author Response · Authors · 2022-11-13
**Thank you for the reviews and revision summary**

We thank all the reviewers for their valuable comments and constructive feedback. We have revised the paper to avoid confusion and reflect your comments (highlighted in blue in the revision). Below is the summary of the revision changes.

1. In the original submission, two learning methods were first presented followed by the experiments. Upon reading some reviews, we find that the current structure could be confusing to identify which experiments are for which methods. Thus we arrange in this order:

   - Sec 3. Method 1: Recurrence transformer with cross-entropy loss (Formerly, Sec 3.1, Sec 3.2)

   -  Sec 4. Experiment to show the advantage of Method 1 in supervised learning (Formerly, Sec 4.1, Sec 4.3): learning constraints from fully labeled data

   -  Sec 5. Method 2 (On top of Method 1): Extend/enhance Method 1 by injecting constraint loss and experiments related to it (Formerly, Sec 3.3, Sec 4.2): learning converges faster with better accuracy and achieves semi-supervised learning

  In other words, in the revision, Sec 3 and 4 do not use constraint loss but learn constraints purely from data and still achieve SOTA performance.

2. We updated tables 7-12 in the appendix by separating the values related to the model structure and the values that are hyperparameters. We want to emphasize that, compared to a standard transformer-based model, our model has only 1 more hyperparameter, which is the number of recurrences.

3. We made the distinction between the original CL-STE [Yang et al., 2022] and our proposed cardinality constraint loss in terms of computation size (to clarify novelty and contrast). we added an ablation study in Appendix B.4 where we applied different losses to the RRN model on SATNet textual Sudoku dataset.

---

### Decision · Program_Chairs · 2023-01-20

**Decision:**

Accept: poster

**Justification For Why Not Higher Score:**

* The proposed architecture is a variant of transformers where recurrence was added.  While this is not particualrly novel, its application to constraint satisfaction problems is novel

**Justification For Why Not Lower Score:**

* Achieves state-of-the-art performance on difficult ungrounded visual sudoku datasets, beating the baseline by a huge margin.
* Generally well-written paper
* Extensive analysis and ablations

**Metareview: Summary, Strengths And Weaknesses:**

The paper proposes a transformer-based arhictecture to learn and solve constraint satisfaction problems

Strengths:
* Achieves state-of-the-art performance on difficult ungrounded visual sudoku datasets, beating the baseline by a huge margin.
* Generally well-written paper
* Extensive analysis and ablations

Weaknesses:
* The proposed architecture is a variant of transformers where recurrence was added.  While this is not particualrly novel, its application to constraint satisfaction problems is novel

This is very interesting work that has led to a significant improvement in the state of the art of ungrounded visual sudoku.

**Note From Pc:**

if the above contains the word "oral" or "spotlight" please see: "oral" presentation means -> notable-top-5% and "spotlight" means -> notable-top-25%. As stated in our emails, we are disassociating presentation type from AC recommendations